# The kinase NEK6 positively regulates LSD1 activity and accumulation in local chromatin sub-compartments
Franziska Knodel [1], Jürgen Eirich [2], Sabine Pinter[1], Stephan A. Eisler[3], Iris Finkemeier [2] &
Philipp Rathert [1] ✉

LSD1 plays a crucial role in mammalian biology, regulated through interactions with coregulators and post-translational modifications. Here we show that the kinase NEK6 stimulates LSD1 activity in cells and observe a strong colocalization of NEK6 and LSD1 at distinct chromatin sub-compartments (CSCs). We demonstrate that LSD1 is a substrate for NEK6 phosphorylation at the N-terminal intrinsically disordered region (IDR) of LSD1, which shows phase separation behavior in vitro and in cells. The LSD1-IDR is important for LSD1 activity and functions to co-compartmentalize NEK6, histone peptides and DNA. The subsequent phosphorylation of LSD1 by NEK6 supports the concentration of LSD1 at these distinct CSCs, which is imperative for dynamic control of transcription. This suggest that phase separation is crucial for the regulatory function of LSD1 and our findings highlight the role of NEK6 in modulating LSD1 activity and phase separation, expanding our understanding of LSD1 regulation and its implications in cellular processes.

The histone lysine demethylase 1 (LSD1 or KDM1A) mediates demethylation of histone H3K4me1/2, thereby conducting a transcriptional repressor function[1–3], partly through downregulation of enhancers[4]. In addition, LSD1 exhibits coactivator activity and has been shown to activate the expression of target genes through histone H3K9 demethylation[2,3,5,6]. In addition to its overarching function as a key transcriptional regulator, LSD1 has been shown to influence processes such as cell fate determination, cell proliferation, and chromosome segregation by either altering the epigenetic landscape of various regulatory genes[7,8] or through the direct demethylation of non-histone proteins, such as transcription factors[9]. Inactivation of LSD1 in mouse embryonic stem cells reveals that it indirectly supports mitotic spindle orientation via β-catenin protein stabilization[10]. Additionally, LSD1-dependent histone demethylation is essential for proper nuclear assembly at the end of mitosis and the timely re-establishment of interphase chromatin[11]. Despite its role in the transcriptional regulation of mitosis-related genes, no direct involvement of LSD1 in mitosis itself has been identified to date. On the contrary, LSD1 is phosphorylated by the polo-like kinase 1 (PLK1) serine/threonine kinase that promotes the release of LSD1 from chromatin during mitosis[12]. Furthermore, LSD1 is a well-documented epigenetic drug target in cancer. High levels of its expression in various tumors are correlated with poor prognosis[13–15]. Knockdown (KD) or inhibition of LSD1 reduces the proliferative capacity of cancer cells in different

tumor models[16,17] and numerous LSD1 inhibitors currently undergo clinical assessment for cancer therapy, particularly for small cell lung cancer (SCLC) and acute myeloid leukemia (AML)[18]. In order to aid the extension of LSD1 inhibitors to other cancer contexts or the development of cotreatment regimes, it is critical to understand the complex levels of LSD1 regulation in more detail. The interaction of LSD1 with coregulatory complexes, such as CoREST, NURD or NCOR[7,19,20], Estrogen Receptor (ER) containing complexes[6], and various transcription factors impose a strong effect on its activity[4]. Additionally, LSD1 activity is regulated through binding to non-coding RNA[21] and the presence of R-loops[22]. Furthermore, many post-translational modification (PTM) sites have been identified on LSD1 preferentially clustering at the N-terminus of the protein[9]. Besides methylation[23], phosphorylation is most prevalent at amino acids located at the N-terminus of LSD1[24] and several kinases have been described, which are responsible for phosphorylation of several serine or threonine residues in this region[9].

Using a fluorescent reporter system to track LSD1 activity combined with a multiplexed RNAi screen we previously identified LSD1 coregulators[22]. In this unbiased and systematic approach, the NIMA-related serine/ threonine kinase NEK6 scored among the top 5 mayor hits positively affecting LSD1 activity. NEK6 was reported to cooperate with NEK7 and NEK9 during mitosis[25]. The Nek6/7/9 signaling module is involved in numerous stages of mitosis and was reported to play a role in centrosome

[1]Department of Biochemistry, Institute of Biochemistry and Technical Biochemistry, University of Stuttgart, Stuttgart, Germany. [2]Institute of Plant Biology and Biotechnology, University of Münster, Münster, Germany. [3]Stuttgart Research Center Systems Biology (SRCSB), University of Stuttgart, Stuttgart, Germany. ✉e-mail: philipp.rathert@ibtb.uni-stuttgart.de

separation and maturation, nuclear envelope breakdown, metaphase and anaphase progression, mitotic spindle formation, and cytokinesis[26–34]. Substrate specificity analysis showed that all three kinases of this mitotic signaling module, Nek6, Nek7, and Nek9, have an almost identical phosphorylation-site motif to the polo-like kinase 1 (PLK1)[35]. Although Nek6 and Nek7 share more than 80% sequence homology in their kinase domain, their mitotic functions are not completely redundant[34]. Their short, disordered N-terminal domains differ significantly and are most likely driving differential interactions or affecting substrate selection, which was observed after removal of the N-terminal domain of NEK6[36,37].

In the recent years, the historical view of chromatin has been challenged by an increasing number of publications, which demonstrate that membrane-less compartments formed through a process called phase separation, are involved in nuclear compartmentalization processes[38,39]. Nuclear compartments, or nuclear bodies, come in various sizes and perform specialized functions essential to diverse cellular processes. For instance, the nucleolus plays a key role in ribosome biogenesis, while constitutive heterochromatin domains and polycomb bodies are associated with heterochromatic regions. Other notable examples include clusters of active RNA polymerase II, and PML body complexes at telomeres. Additionally, Cajal bodies and paraspeckles are involved in the biogenesis, maturation, and recycling of small RNAs, as well as RNA editing. Nuclear speckles and stress bodies contribute to the storage and recycling of splicing factors, particularly during cellular stress[40]. These structures are often referred to as chromatin sub-compartments (CSCs), highlighting their involvement in chromatin organization and regulation[41]. The compartmentalization process in general is driven by multivalent interactions of nucleic acids and/or proteins[42–44] in concert with intrinsically disordered regions (IDRs). Proteins involved in this condensation process can be characterized as scaffold or client molecules, where scaffolds drive the actual condensation and clients merge together with the scaffolds without strong intrinsic phase separation tendency. In addition, regulators that can modulate condensation behavior like post-translational modifications have been added to the client and scaffold model[45]. In particular phosphorylation of IDRs plays an important role in the assembly or dissociation of such a condensate[46]. However, the definition of client, scaffold, and regulator is flexible and can overlap occasionally[45]. Biomolecular condensation is further challenged as droplets are reported to age or to mature to solid-like or gel-like structures over time[47–49]. In contrast to NEK6, LSD1 contains a rather long N-terminal IDR and recent publications showed that LSD1 is involved in the formation of distinct nuclear phases in complex with other proteins[50] or lncRNA[51].

Here we show that NEK6 is a critical coregulator of LSD1 activity that phosphorylates LSD1 within its N-terminal IDR at serine 126 (S126). The phosphorylation and the interaction with NEK6 lead to the establishment of distinct CSCs in part driven by phase separation mechanisms, where NEK6 serves as the scaffold that recruits LSD1. The subsequent phosphorylation of LSD1 serves as a positive feedback loop during the establishment and maintenance of distinct subcellular compartments thereby positively influencing LSD1 activity at specific genomic regions.

## Results

### NEK6 modulates LSD1 activity in cells
In a recent publication, we identified novel functional coregulators of LSD1 in a comprehensive fashion using a fluorescent reporter system monitoring the activity of LSD1 in living cells combined with a functional genetic screening approach[22]. Among the 5 top scoring hits we identified the NIMA-related Kinase 6 (NEK6)[22], which was not reported to be connected to LSD1 activity before. Initial validation using this reporter system to monitor LSD1 activity in living cells (Fig. 1a) confirmed the screening results and demonstrated that LSD1 activity was reduced (Fig. 1b) after suppression of NEK6 expression (Supplementary Fig. 1a). Therefore, we set out to understand the role of NEK6 in the regulation of LSD1 biology in more detail. So far, NEK6 has only been described as being involved in mitotic phosphorylation events and therefore necessary for successful mitotic

progression[26,34,35,52]. Furthermore, NEK6 was reported to play a role in several signaling pathways[53–55] and DNA damage response in prostate cancer[56,57]. To further understand how NEK6 influences LSD1 activity we investigated the colocalization of both endogenous proteins in a mouse fibroblast (NIH/3T3) and human embryonic kidney (HEK293) cell line using immunofluorescence microscopy. LSD1 showed a dispersed localization throughout the nucleus with few discrete foci per nucleus. Interestingly, we observed a striking colocalization of LSD1 with NEK6 within these distinct nuclear foci, which formed outside of heterochromatic regions as visualized by DAPI staining (Fig. 1c). Despite the reported function of NEK6 in mitosis we did not observe specific colocalization of LSD1 and NEK6 in mitotic cells, suggesting that the function of NEK6 on LSD1 activity is independent of cell cycle regulation (Supplementary Fig. 1b and c).

Taken together, we demonstrate that suppression of NEK6 negatively affects LSD1 activity in cells using the fluorescent reporter system and we observe a profound colocalization of both proteins in different mouse and human cell types.

### NEK6 phosphorylates LSD1 at serine 126
Due to the clear colocalization of both proteins and the effect on LSD1 activity upon NEK6 suppression, we hypothesized that LSD1 might be a substrate of NEK6 and incubated recombinant LSD1 and NEK6 (Supplementary Fig. 2a) in the presence of radiolabeled ATP. This showed a clear phosphorylation signal at the expected size of LSD1 (Fig. 2a). Phosphorylation of recombinant Histone 3.1 (H3.1) by NEK6, which was reported previously[52] and was included as a positive control, showed a much weaker signal (Fig. 2a). To identify the potential site of LSD1 phosphorylation the recombinant proteins were incubated and the reaction was analyzed by liquid chromatography–mass spectrometry (LC-MS/MS). The quenched reaction was digested with three different peptidases (Trypsin, GluC, and ProAlanase) in order to maximize a sequence coverage. We detected 94% of the amino acid sequence of LSD1 allowing to capture 116 S, T, and Y amino acid residues as potential phosphorylation sites, except T783 and S787 (Supplementary Table 1). In general, three different residues appeared to be phosphorylated: S126, T500, and T803 (Fig. 2b, c and Supplementary Fig. 2d, e). However, T500 showed phosphorylation without the presence of NEK6 (Supplementary Fig. 2d) and we exchanged all potential target sites with alanine (A) in the recombinant LSD1 construct. This led to a significant reduction of the phosphorylation signal upon substitution of S126 with A (S126A) (Fig. 2d). This effect was not observed when T500 and T803 were exchanged with A (Supplementary Fig. 2e) indicating that S126 is the major target for phosphorylation by NEK6. This effect could be further enhanced by performing similar experiments with the disordered N-terminal domain of LSD1 (Fig. 2e, Supplementary Fig. 2c), which confirmed the LC-MS/MS results showing that NEK6 is able to phosphorylate different residues of LSD1 in vitro with a preference for S126. The LSD1 region containing the phosphorylated residue is highly conserved across different vertebrates (Fig. 2f) suggesting that this region might play an important role in regulating LSD1 activity. However, the complete region is absent in phylogenetically more distant species such as C. elegans and D. melanogaster.

Taken together, we could demonstrate that NEK6 phosphorylates LSD1 at S126, a residue located in a highly conserved protein region.

### The N-terminus of LSD1 is disordered and exhibits phase-separation behavior in vitro
Serine 126 has already been reported to be phosphorylated by PLK1 in a cell cycle-dependent manner leading to a displacement of LSD1 from chromatin[12]. However, we did not observe a distinct colocalization of NEK6 and LSD1 in cells during different stages of mitosis. In fact, the profound colocalization of respective proteins was absent in mitotic cells (Supplementary Fig. 1b and c), which suggested that NEK6 phosphorylation of LSD1 is not cell cycle-dependent and most likely has a different function. Interestingly, the N-terminus of LSD1 is highly disordered based on different prediction tools (Fig. 3a, Supplementary Fig. 3a). Initial investigation

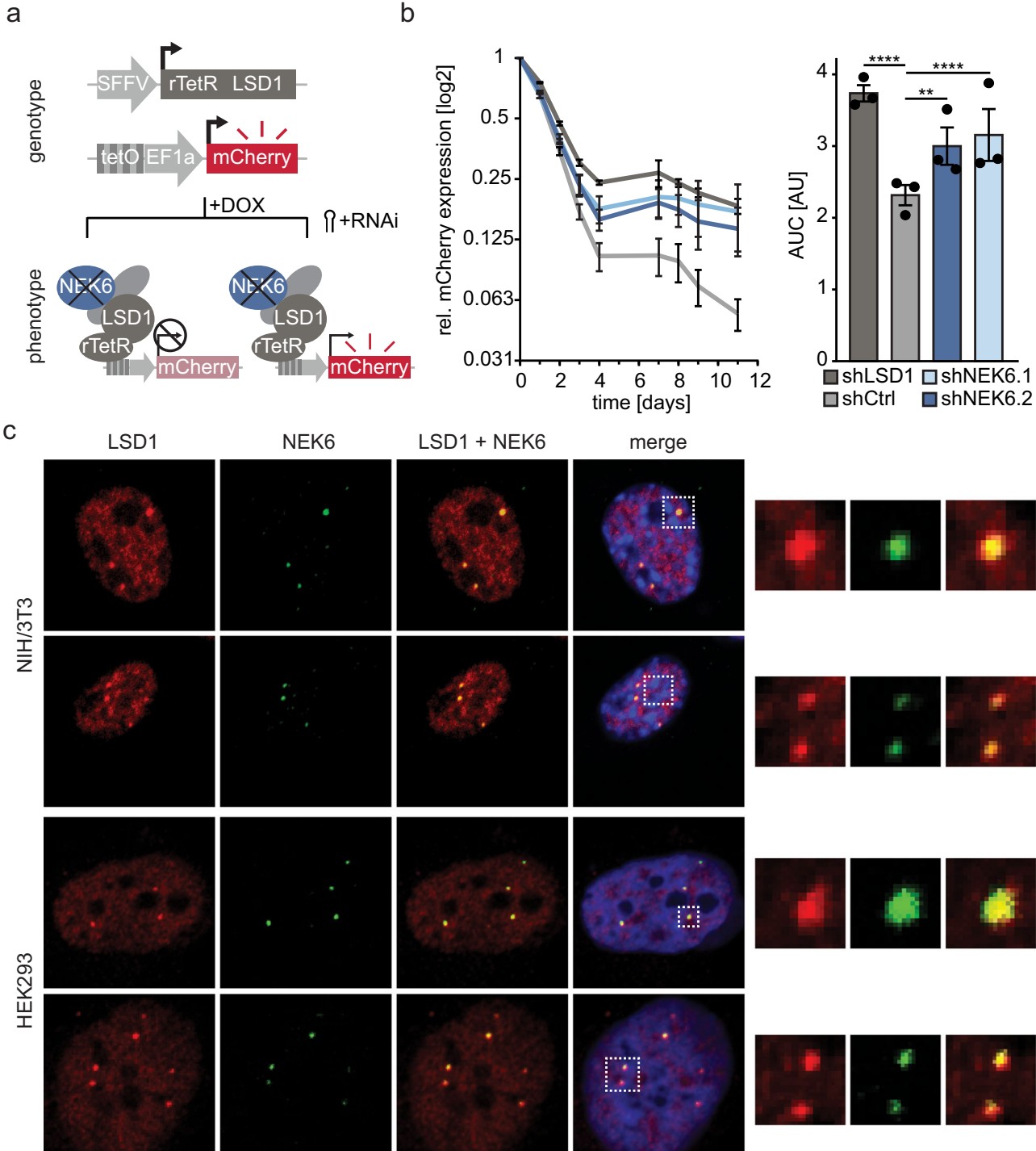

**Fig. 1 | NEK6 modulates LSD1 activity in cells. a** Illustration depicting the principle of the fluorescent reporter system. Cells of interest are transduced with a vector expressing the fluorescent reporter under control of the synthetic promoter. Subsequent transduction with a construct expressing the fusion protein of human LSD1 and the reverse tetracycline repressor protein (rTetR), both with selection cassettes and expression under a constitutive promotor enables the observation of LSD1 activity in living cells. Upon doxycycline (DOX) treatment, the rTetR-LSD1 fusion protein is recruited to the synthetic promoter, together with endogenous co-regulators, leading to the suppression of mCherry expression. Interference with the expression of LSD1 co-regulators (e.g., NEK6) influences LSD1 silencing activity. **b** Left: Validation of the coregulatory effect of NEK6 on LSD1. Reporter fluorophore expression was monitored over the course of 11 days in NIH/3T3 cells expressing the

following shRNAs: shNek6.1, shNek6.2, shLsd1 (positive control), or shCtrl (neutral control). The median reporter expression of cells with Dox-induced rTetR-LSD1 recruitment was normalized to reporter cells without rTetR-LSD1 recruitment. The x-axis shows the time course. Right: Statistical analysis was performed by a one-way ANOVA of the area under curve followed by Dunnett´s multiple comparisons test ($n = 3$, mean ± SEM; ****:$p < 0.0001$, **:$p < 0.001$). **c** Representative immuno-fluorescence microscopy images of NIH/3T3 cells (top) and HEK293 cells (bottom) stained for endogenous LSD1 and NEK6 show clear colocalization in spherical nuclear sub-compartments. Nuclei were stained with DAPI. Zoom-in pictures of respective colocalized spots marked with boxes are shown for LSD1, NEK6, and for merged channels. Scale bars = 10 µm.

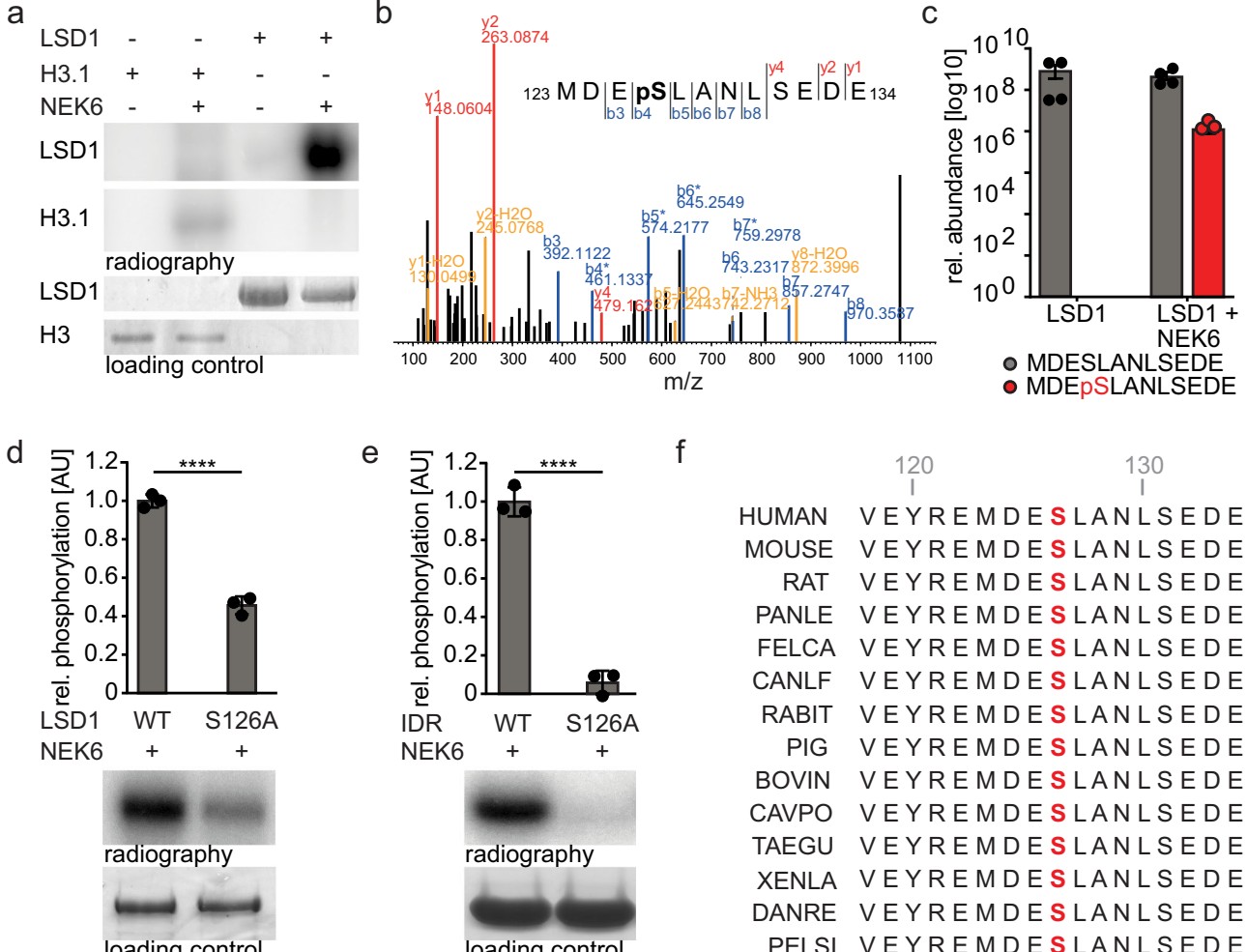

**Fig. 2 | NEK6 phosphorylates LSD1 at serine 126 (S126). a** Representative LSD1 and H3.1 phosphorylation signal measured by autoradiography. His-tagged LSD1 and histone H3.1 protein were incubated with GST-NEK6 in presence of radiolabeled [γ-$^{32}$P]-ATP and separated on a 12% SDS-polyacrylamide gel. Phosphorylation was detected by autoradiography. A Coomassie Brilliant Blue stained 12% SDS-polyacrylamide gel served as a loading control. **b** Annotated MS$^2$ fragmentation spectrum of MDEpSLANLSEDE with the identified fragment ions and respective start and end positions of the peptide within LSD1 indicated additionally in the inset. b ions blue, y ions red, fragment ions with losses yellow. **c** Intensity average of the respective LSD1 peptides from 4 LC-MS replicates. **d** Quantification of LSD1 phosphorylation signal measured by autoradiography. Equal amounts of His-tagged wt LSD1 FL or LSD1 FL S126A were incubated with GST-NEK6 in presence of radio

labeled [γ-$^{32}$P]-ATP and separated on a 12% SDS-polyacrylamide gel. Phosphorylation was detected by autoradiography (bottom). A Coomassie Brilliant Blue stained 12% SDS-polyacrylamide gel served as a loading control. ($n = 3$, mean ± SEM, unpaired t-test, ****:$p < 0.0001$). **e** Quantification of LSD1 phosphorylation signal measured by autoradiography. Equal amounts of His-tagged WT LSD1 N-terminal domain (IDR, AA 1-191) or IDR S126A were incubated with GST-NEK6 in presence of radiolabeled γ-32P-ATP and separated on a 12% SDS-polyacrylamide gel. Phosphorylation was detected by autoradiography (bottom). A Coomassie Brilliant Blue stained 12% SDS-polyacrylamide gel served as a loading control. ($n = 3$, mean ± SEM, unpaired t-test, ****:$p < 0.0001$). **f** Protein alignment of the amino acids 117-134 of human LSD1 and 14 indicated species. Phosphorylation target S126 is marked in red.

of the recombinant LSD1 IDR using circular dichroism measurements did not show any correlation with the reference data of secondary structure elements (Fig. 3b). The IDR of LSD1 contains a large percentage of hydrophobic amino acid residues followed by polar residues equally distributed across the IDR sequence and various acidic and basic patches (Supplementary Fig. 3b). Previous studies have implicated low-complexity IDRs of proteins in liquid-liquid phase separation[43,58–61] and phosphorylation sites are located predominantly in such regions[62]. The addition of a negatively charged phosphate moiety alters the physicochemical nature of a protein region and phosphorylation of serine and threonine residues was reported to affect phase separation properties of the modified proteins[63], which was already demonstrated for other nuclear proteins[38,50,64]. Besides this, LSD1 has been reported recently to participate in LLPS formation of ZYMDN8[50,51]. Inspired by these data we cloned and purified the IDR of LSD1 fused to mVenus (IDR-mV) for further in vitro characterization

(Supplementary Fig. 3c). As anticipated, we observed the development of a turbid solution upon salt reduction and simultaneous addition of a crowding agent in vitro, but not by mV alone (Fig. 3c). Fluorescence microscopy revealed the formation of spherical condensates (Fig. 3c), which increased in size with increasing protein concentration (Fig. 3d). Droplet formation was largely unaffected by different salt concentrations ranging from 30 to 100 mM KCl as well as different PEG concentrations (Supplementary Fig. 3d, e), but strongly influenced by pH alterations (Fig. 3e). Additionally, droplet coalescence was observed, which led to an increase in droplet area after the fusion event (Fig. 3f and g, Supplementary videos 1–3), confirming the dynamic nature of the droplets. Additional fluorescence recovery after photobleaching (FRAP) experiments revealed a strong and fast fluorescence recovery at the bleached area (Fig. 3h) demonstrating the liquid and mobile behavior of the IDR condensates. To determine if the LSD1 IDR droplets exhibit aging characteristics in which they transition

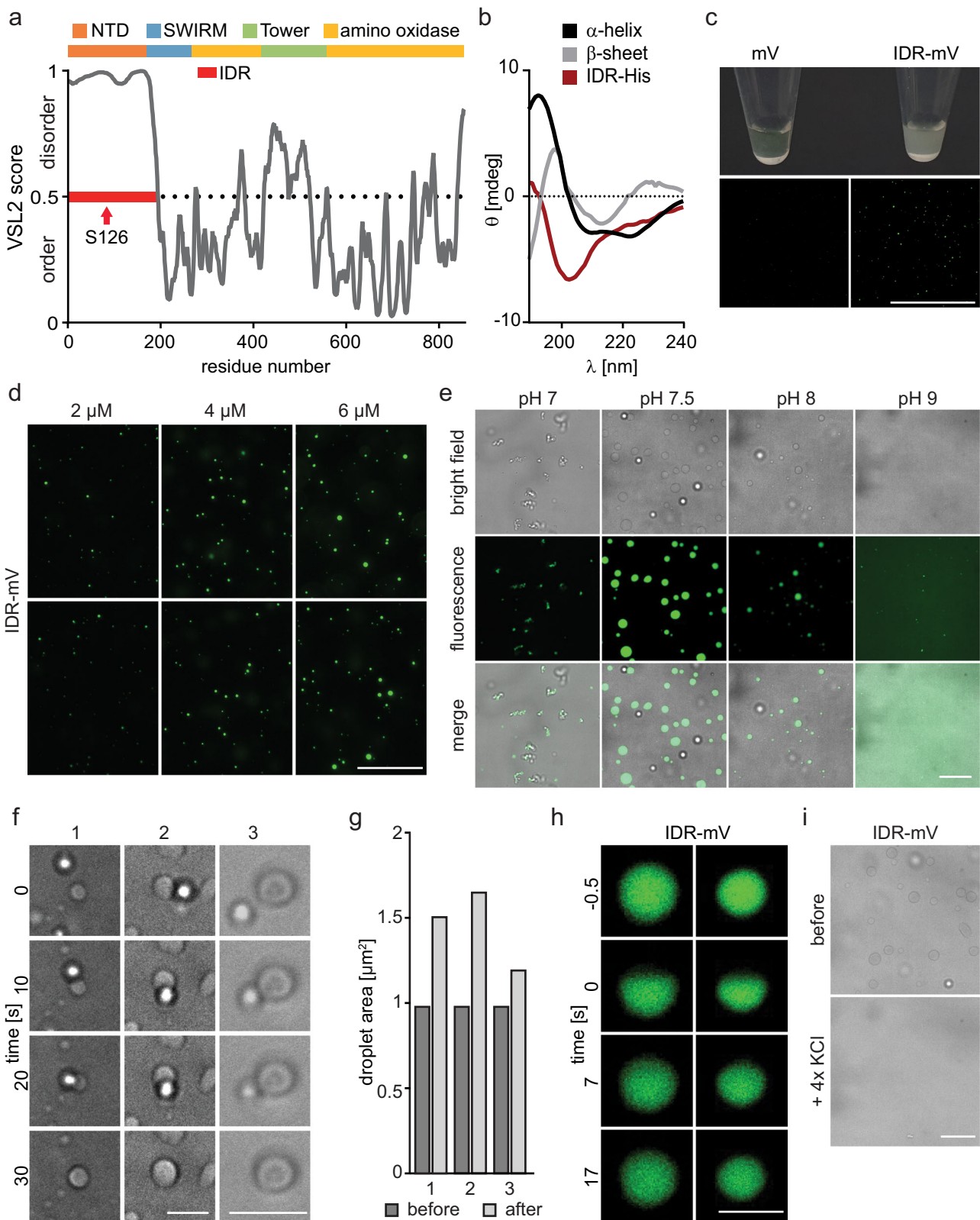

from liquid-like to a more solid-like state we determined the reversibility of droplet formation through the addition of KCl and monitored the dissolution of droplets by the addition of salt (Fig. 3i).

This initial characterization demonstrates that the N-terminus of LSD1 is unstructured and shows typical characteristic for biomolecular condensates in vitro.

## Phosphorylation of S126 affects phase separation

Based on the knowledge that phosphorylation can affect phase separation properties of proteins[46] we hypothesized that the phosphorylation at S126 may alter the phase separation properties of the IDR and measured the turbidity at $OD_{600}$ of a recombinant serine to glutamate (S126E) IDR mutant, which is generally used to mimic a specific phosphorylation. The

**Fig. 3 | The N-terminus of LSD1 is disordered and shows phase separation behavior. a** Intrinsic disorder prediction of LSD1 using the PONDR VSL2 algorithm[108]. The N-terminal domain (NTD) designates the IDR containing the phosphorylation site under investigation. Additional domains of LSD1 are indicated[109]. The X-axis depicts residue numbers plotted against the degree of folding prediction. **b** CD spectra of the LSD1 IDR-His compared to reference data of secondary structure elements for α-helix and β-sheet[110]. **c** Phase separation of purified IDR-mV compared to mV (10 μM). Phase separation was induced by the addition of 20% PEG and reduction of KCl to 10 mM. Top: macroscopic phase separation of IDR-mV. Bottom: fluorescence images showing condensed droplets of IDR-mV compared to mV, Scale bar = 200 μm. **d** Representative fluorescence microscopy images showing in vitro droplet formation of IDR-mV under different protein concentrations (2, 4, 6 μM). Scale bar = 50 μm. **e** Representative bright field and fluorescence microscopy images showing in vitro condensation of IDR-mV under different pH conditions. Scale bar = 10 μm. **f** Representative bright-field microscopy images of phase-separated IDR-mV droplets showing a time course of condensate fusion. Scale bar = 5 μm. **g** Size analysis of the droplets shown in (**f**) before and after the fusion event. **h** Representative fluorescence microscopy images at different time points during FRAP experiments proving protein mobility within droplet. Scale bars = 2 μm. **i** Representative bright-field microscopy images of in vitro phase separated IDR-mV droplets before and after addition of external salt showing droplet dissolution. Scale bar = 10 μm.

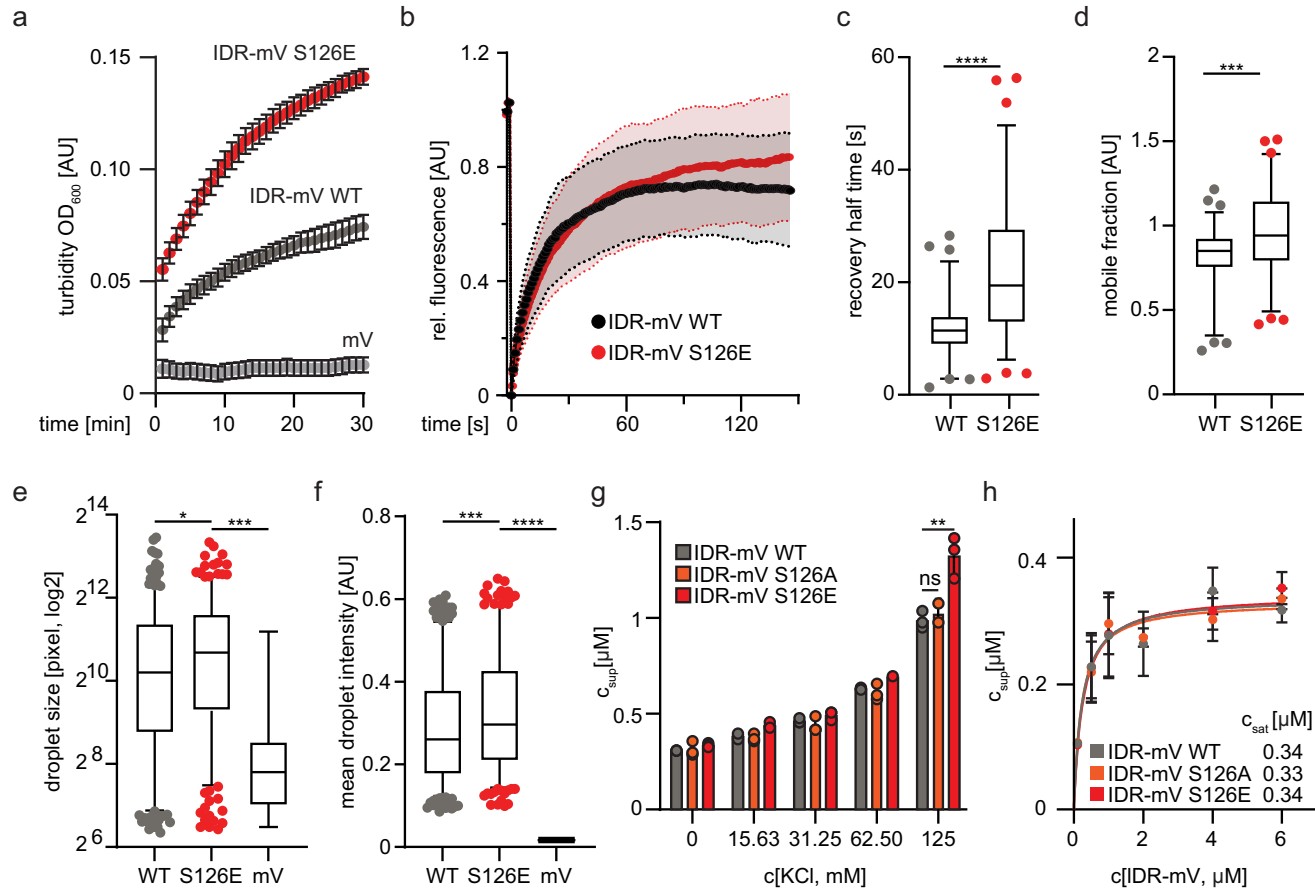

**Fig. 4 | Phosphorylation of S126 influences the phase separation behavior of the LSD1 IDR. a** Turbidity measurements at $OD_{600}$ comparing the optical density increase of IDR-mV S126E with IDR-mV WT. ($n = 3$, mean ± SEM). **b** Quantification of partial droplet fluorescence recovery after photobleaching (FRAP) over time for IDR-mV WT and S126E. ($n = 2$ biological replicates with $N_{WT} = 72$ and $N_{S126E} = 75$ individual bleach events, error bars = median ± SD). **c** Box plots depicting of fluorophore recovery half-time during FRAP experiments of IDR-mV WT vs S126E. ($n = 2$ biological replicates with $N_{WT} = 72$ and $N_{S126E} = 75$ individual bleach events, box plot: 5-95 percentile; mean as center line, ****:$p < 0.0001$; unpaired t-test). **d** Box plots depicting fluorophore mobile fraction during FRAP experiments of IDR-mV WT compared to S126E. ($n = 2$ biological replicates with $N_{WT} = 72$ and $N_{S126E} = 75$ individual bleach events, box plot: 5-95 percentile; mean as center line, ***:$p < 0.001$; unpaired t-test). **e** Box plot depicting the size of phase-separated IDR-mV WT vs S126E droplets. Pictures were analyzed with Cell Profiler. ($n = 2$ with $N_{S126E} = 308$, $N_{WT} = 363$,

$N_{mV} = 15$ mV, box plot: 5-95 percentile; mean as center line, *:$p < 0.1$, ***:$p < 0.001$; ordinary one-way ANOVA followed by Dunnett's multiple comparison). **f** Box plot showing the mean droplet intensity of phase-separated IDR-mV WT vs S126E condensates. Pictures were analyzed with Cell Profiler. ($n = 2$ with $N_{S126E} = 308$, $N_{WT} = 363$, $N_{mV} = 15$, box plot: 5-95 percentile; mean as center line, ***:$p < 0.001$, ****:$p < 0.0001$, ordinary one-way ANOVA followed by Dunnett's multiple comparison). **g** Bar graphs depicting the reversibility of droplet formation. The dilute phase concentration ($c_{sup}$) of the indicated LSD1 IDR variants was determined after addition of increasing KCl concentrations to determine alterations in droplet aging. ($n = 3$, mean ± SEM; **:$p < 0.01$, ordinary one-way ANOVA of data at 125 mM KCl followed by Dunnett's multiple comparison). **h** Quantification of dilute phase concentration ($c_{sup}$) of IDR-mV WT compared to S126A/E. The saturation concentration ($c_{sat}$) was determined by increasing concentrations of the respective LSD1 variants. ($n = 3$, mean ± SEM).

S126E mutant demonstrated a higher and faster increase in turbidity compared to the wild-type (WT) control (Fig. 4a). However, the phosphorylation did not extensively affect the mobility of the molecules within the droplet itself during FRAP recovery (Fig. 4b) and we noticed mild but significant effects with recognizable variance in the recovery half time and the mobile fraction (Fig. 4c, d). In line with these results, the droplet size and intensity were increased in droplets harboring the S126E mutation in contrast to droplets formed by the WT-IDR (Fig. 4e, f). Despite the observed effects with the IDR alone, we did not observe strong consequences of the S126E phosphomimic variant or the S126A mutation, which also shows

phase separation behavior in vitro (Supplementary Fig. 3f), on LSD1 activity in the cellular reporter system or in vitro demethylation kinetics (Supplementary Fig. 3g, h). Interestingly, we observed higher droplet dissolution fraction upon addition of salt for the S126E mutant compared to the LSD1 WT and S126A mutant (Fig. 4g), whereas the saturation concentration ($c_{sat}$) remained unaffected (Fig. 4h).

Our data indicates that the substitution of S126 with E to mimic a phosphorylation by NEK6 affects the phase separation properties of the LSD1 IDR in vitro leading to faster droplet formation, higher mobility, and larger droplets as well as a reduced droplet aging behavior.

### Phase separation promotes local LSD1 and NEK6 association in vitro

NEK6 suppression negatively affects LSD1 activity in vitro and we observed a distinct colocalization of both proteins in local LSD1 and NEK6 CSCs. NEK6 also contains a small N-terminal IDR (Supplementary Fig. 4a), which gave rise to the hypothesis that the interaction of NEK6 and LSD1 leads to the formation of localized CSCs through a phase separation mechanism. In order to investigate if the colocalization of the respective proteins could be in fact guided by phase separation, we induced droplet formation of the IDR-mV in the presence of recombinant NEK6 fused to monomeric dsRed (Supplementary Fig. 4b). We recognized the co-compartmentalization of NEK6-dsRed and IDR-mV (Fig. 5a), whereas dsRed alone did not co-compartmentalize in IDR-mV condensates (Fig. 5b). This implies that the formation of the LSD1-NEK6 CSCs could be mediated through heterotypic interactions between the LSD1 IDR and NEK6. NEK6-dsRed alone formed solid-like aggregated structures, pointing towards the conclusion that the concentration of NEK6 exceeds its solubility (Supplementary Fig. 4c) within this experiment, but is soluble in combination with the LSD1 IDR. Treatment of mouse fibroblast cells with 10% 1,6-Hexanediol for 60 seconds, a widely used but debated tool to probe LLPS in cells[65] that disrupts hydrophobic interaction-induced phase separation assemblies[66] led to a complete loss of the LSD1 and NEK6 CSCs (Supplementary Fig. 4d, e), which is an indication that LSD1-NEK6 CSCs are formed in cells by mechanisms observed in biomolecular condensates.

### LSD1-NEK6 CSC establishment is impacted by phase separation

To further investigate the impact of the IDR of LSD1 on the establishment of localized CSCs, we aimed to determine the localization of ectopically expressed NEK6, the full-length (FL-) catalytically inactive K661A LSD1 mutant and the LSD1 IDR fused to mV in mouse fibroblast cells, which demonstrated equal distribution within the nucleus (Supplementary Fig. 4f). This suggests that induction of condensation at particular subcellular locations is ineffective due to various reasons such as disproportionate high concentration of the ectopically expressed protein or a dilute concentration of other ligands important for the establishment of such compartments. For example, RNA transcripts associated with specific DNA loci[67–69] as well as DNA[70–72] function as an oligomerization platform that facilitates the formation of CSCs locally enriching self-interacting IDRs[41,73]. To overcome such problems and to rapidly and precisely induce condensate formation at particular subcellular locations we applied the Corelet system[74], which serves as model system for condensates. This approach makes use of two modules: a GFP-tagged "Core" comprized of 24 human ferritin heavy chain (FTH1) protein subunits, which are functionalized with the improved light-inducible dimer (iLID) domain and the cognate binding partner (SspB) fused to mCherry and the protein of interest (POI), e.g., LSD1 or NEK6. To mimic the native oligomerization process, blue light is applied to induce the binding between iLID and SspB. The presence of a protein or an IDR able to induce phase separation generates microscopically detectable liquid droplets. In response to blue light we monitored a defined condensation for FL-NEK6 and catalytically inactive K661A FL-LSD1 after fusion to SspB (Fig. 5c; Supplementary videos 4 and 5) suggesting that effective oligomerization appears to be necessary for the establishment of distinct CSCs. The observed formation of nuclear Corelet compartments showed to be

reversible and we monitored decondensation upon removal of blue light (Supplementary videos 8 and 9), suggesting mobile and reversible condensate behavior in cells.

Interestingly, the NEK6 Corelet condensates appeared more rapidly, showed a more spherical appearance, and demonstrated higher mobility in contrast to the LSD1 Corelet condensates (Fig. 5c and Supplementary videos 6 and 7), which is supported by the observation that NEK6 condensates exhibited a liquid-like coalescence behavior in cells (Fig. 5d). A phenomenon we did not observe with the FL-LSD1 Corelets, which show irregular morphology and uneven boundaries. This could be a consequence of the multivalent nature and the resulting robust association of LSD1 to nuclear loci. We sequentially truncated the N-terminus of LSD1 and observed that FL-LSD1 is required for Corelet droplet formation. Neither the IDR alone nor any truncation showed effective formation of condensates in cells (Supplementary Fig. 5a) suggesting that the establishment of LSD1 CSCs observed with the enogenous LSD1 (Fig. 1c, Supplementary Fig. 4d,e) might rely on multivalent protein interactions, in contrast to the in vitro data implying phase separation mediated by the IDR. This hypothesis is supported by the observation that the LSD1 IDR did incorporate different histone peptides harboring various modifications (Fig. 5e and Supplementary Fig. 4g) and DNA (Fig. 5f). On the other hand, removal of the IDR of NEK6 did not alter the condensation behavior whereas the IDR of NEK6 alone was ineffective in Corelet droplet formation (Fig. 5g).

These results highlight the complex interplay of in vitro biochemical activity, condensate formation, and cellular function. The formation of cellular CSCs is not only dependent on the observed in vitro phase separation but depends on various additional interactions and oligomerization most likely induced by the multivalency of LSD1 showing interaction with chromatin components (DNA, histones and RNA) as well as additional coregulators and complex partners. Our data suggest that NEK6 is an important factor in this oligomerization process and shows phase separation behavior in cells, which is highly mobile and independent of its IDR.

### Subcellular co-localization of LSD1 and NEK6 is guided by the LSD1 IDR

We further examined the contribution of the IDRs of LSD1 and NEK6 on the recruitment of the respective partner[75]. Initially, we probed if the endogenous behavior of LSD1 and NEK6 forming specific CSCs is recapitulated in the Corlet system and fixed cells with LSD1 or NEK6 Corelets after light activation and performed immunofluorescence staining for the respective endogenous partner. Upon long-term blue light activation, we observed a distinct colocalization of NEK6 Corelets with endogenous LSD1 and vice versa (Fig. 5h). To assure that this incorporation is indeed mediated by the respective interaction partner we stained for LSD1 and NEK6 in FUS-Corelet condensates, but did not observe such profound colocalization, neither for LSD1 nor for NEK6, indicating that the interaction between NEK6 Corelet and endogenous LSD1 is indeed based on a direct interaction (Supplementary Fig. 5b, c). We further analyzed if the LSD1 presence within NEK6 Corelet droplets is influenced by NEK6-IDR and did not observe any changes in co-compartmentalization of endogenous LSD1 in Δ35-NEK6 Corelets (Fig. 5h) suggesting that the incorporation of LSD1 into NEK6 CSCs is independent on the IDR of NEK6. Intrigued by our results and the importance of the LSD1 IDR for the formation of LSD1 CSCs using the Corelet system, we aimed to determine the direct influence of the IDR on LSD1 activity. To this end we used the fluorescent reporter system (Fig. 1a) to analyze the silencing capacity at the reporter gene of FL-LSD1 in comparison to shortened LSD1 variants harboring N-terminal truncations (Fig. 5i and Supplementary Fig. 5d). Strikingly, a gradual reduction of reporter silencing was observed correlating with the degree of truncation at the N-terminus of LSD1. The Δ60 variant, lacking the first 60 amino acids exhibits comparable activity as the FL enzyme, but larger truncations show a significant and gradual decrease in silencing of the fluorescent reporter gene.

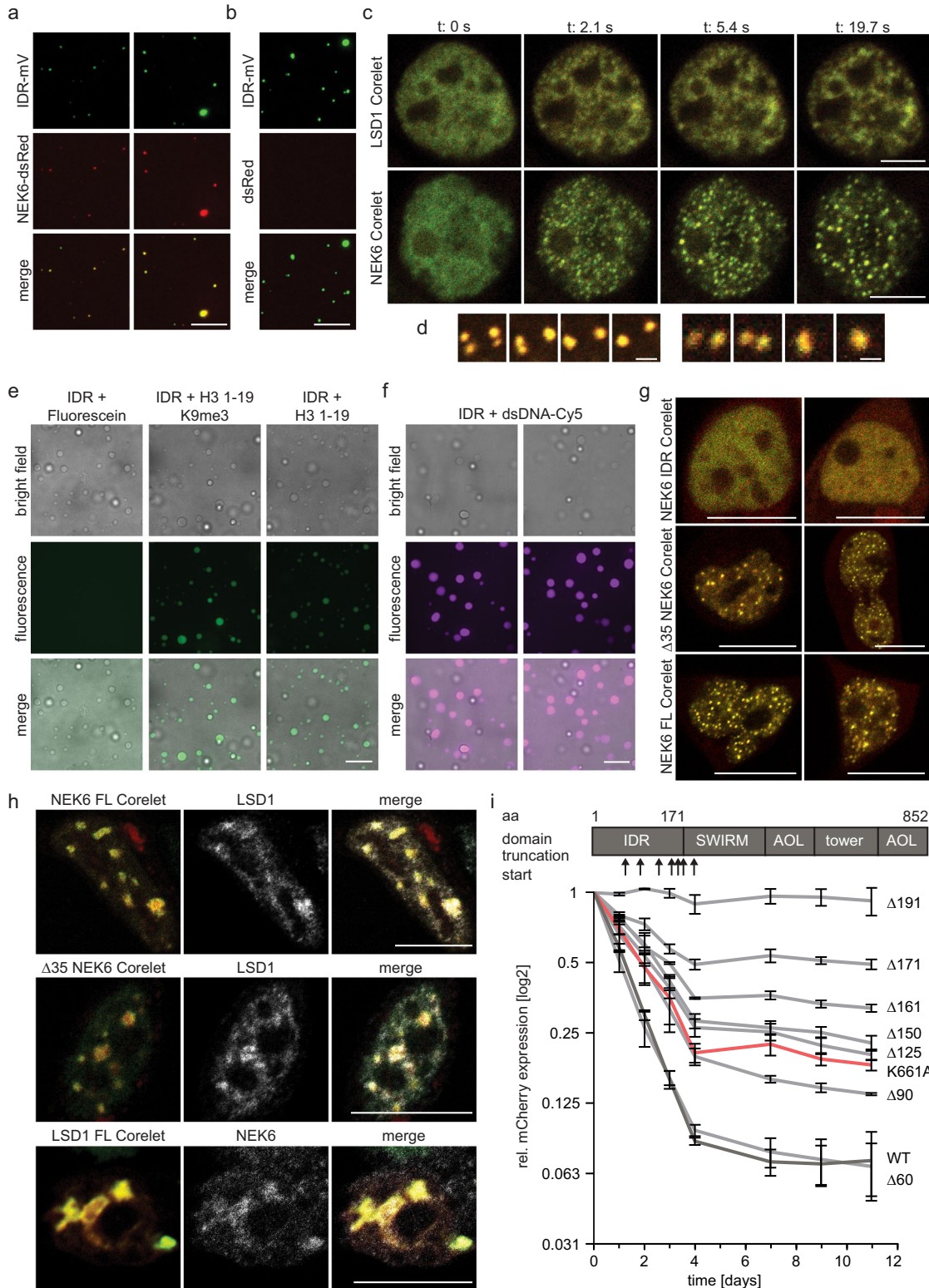

The truncation lacking the first 191 amino acids (Δ191) shows no activity, which was anticipated because at least 20 amino acids of the SWIRM domain are removed as well, potentially resulting in a misfolded catalytic domain of LSD1 (Fig. 5i and Supplementary Fig. 5d).

This demonstrates that the LSD1-NEK6 CSC formation is dependent on the presence of the LSD1 IDR, but does not necessarily require the IDR of NEK6. Furthermore, the silencing efficiency in cells is greatly dependent on the integrity of LSD1 IDR.

## Discussion

LSD1 is a major chromatin regulator and important drug target in different tumor types, which has been demonstrated to localize to actively transcribed genomic regions[1] where its enzymatic activity is suppressed due to different proposed mechanisms[1,76]. LSD1 is activated when transcribed regions must be silenced e.g., during normal differentiation. This process requires the presence of additional coregulator proteins and the kinase NEK6 was found recently to positively influence LSD1 silencing activity in living cells[22]. We

**Fig. 5 | Phase separation promotes local LSD1 and NEK6 association.**
**a** Representative fluorescence microscopy images depicting in vitro co-condensation of IDR-mV and NEK6-dsRed. 2 μM of both proteins was incubated under crowding conditions and transferred onto glass slides for microscopy. Scale bar = 10 μm.
**b** Representative fluorescence microscopy images of in vitro IDR-mV droplets showing no incorporation of dsRed. 2 μM of both proteins was incubated under crowding conditions and transferred onto glass slides for microscopy. Scale bar = 10 μm. **c** Representative live cell fluorescence microscopy images of HEK293 cells expressing either LSD1 K661 or NEK6 Corelet during blue light activation. Structures formed by LSD1 reflect less mobility and less round shape in comparison to NEK6 Corelet. Scale bar = 5 μm. **d** Fusion event during Corelet phase separation induction of NEK6. Scale bar = 2 μm. **e** Representative fluorescence and bright field microscopy images demonstrating strong incorporation of FITC-tagged histone H3.1 peptides (peptide length as indicated) within phase-separated droplets formed by IDR-His. Scale bar = 10 μm. **f** Representative fluorescence and bright field microscopy images demonstrating strong incorporation of Cy5-labeled dsDNA

within phase-separated droplets formed by IDR-His. Scale bar = 10 μm.
**g** Representative live cell fluorescence microscopy images of HEK293 cells expressing different NEK6 Corelet constructs during blue light activation. NEK6-IDR is insufficient for Corelet droplet formation, whereas Δ35-NEK6 shows comparable Corelet condensation upon blue light as FL-NEK6. Scale bar = 5 μm.
**h** Representative immunofluorescence images of HEK293 cells expressing FL-NEK6, Δ35-NEK6 or FL-LSD1 K661A with complementary endogenous protein staining. Corelet condensation was induced for 4 h with 5 min on/off cycles and proteins were fixed during blue light emission. Both proteins are capable to concentrate the respective partner into Corelet droplets. Scale bar = 10 μm. **i** Median reporter expression during the recruitment of rTetR-LSD1 WT and rTetR-LSD1 K661A in comparison to the indicated rTetR-LSD1 truncations as depicted on top. Upon shortening of the IDR, the silencing capacity of LSD1 is gradually reduced. The median reporter expression of cells with Dox-induced rTetR-LSD1 recruitment was normalized to reporter cells without rTetR-LSD1 recruitment. The x-axis shows the time course.

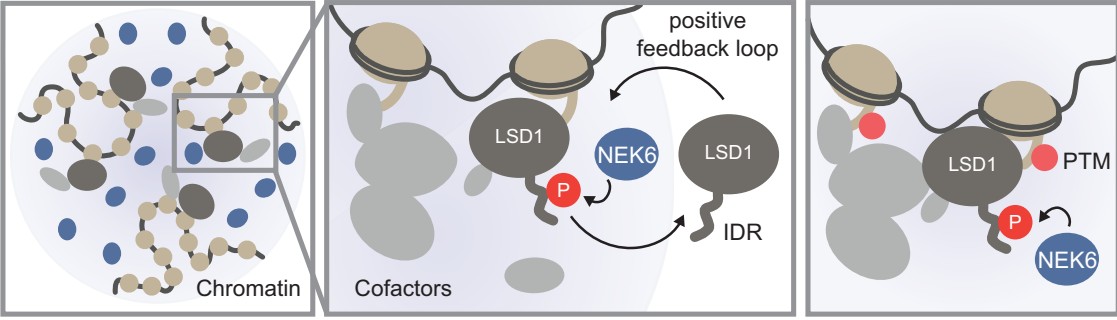

**Fig. 6 | Local NEK6-LSD1-CSCs impact LSD1 activity.** The Ser/Thr kinase NEK6 colocalizes with LSD1 in specific chromatin sub-compartments (CSCs). The subsequent phosphorylation of LSD1 functions as a positive feedback loop leading to an increase in local substrate-enzyme-cofactor concentration and faster demethylation rates. This contributes to the establishment of multidimensional silencing hubs crucial for robust gene repression.

show now that NEK6 improves LSD1 activity through its contribution in the establishment of nuclear CSC[41], in which NEK6 and LSD1 are concentrated in distinct nuclear foci. Within these CSCs NEK6 functions as a positive regulator of LSD1 activity with downstream implications for gene expression (Fig. 6).

In the present study, we show that NEK6 suppression negatively affects LSD1 activity using a synthetic reporter system (Fig. 1). Interestingly, LSD1 is phosphorylated by NEK6 in vitro at S126, which is located within the N-terminal IDR of LSD1 (Fig. 2). This region is highly conserved in vertebrate organisms, but not present in non-vertebrate model organisms such as C. elegans or D. melanogaster.

It was recently shown that PLK1 phosphorylates LSD1 at the same residue leading to the release of LSD1 from chromatin during mitosis[12]. However, we did not observe that the phosphorylation negatively affects the enzymatic activity of LSD1 (Supplementary Fig. 3g) and the silencing activity in cells (Supplementary Fig. 3h). In contrast, our data indicates that efficient silencing of LSD1 in cells depends on the presence of NEK6 and the phosphorylation of S126, mimicked with the substitution of S to E, which enhances the phase separation propensity of the IDR in vitro (Fig. 4). We hypothesize that this effect is most likely also occurring within the CSC and raised a S126-phosphospecific antibody, which was ineffective in detecting LSD1 phosphorylation in immunofluorescence experiments and prevented us from providing this data. Despite the fact that we did not detect any effects of the phosphomimic variant in our cellular reporter assay, we cannot exclude that the phosphorylation at S126 affects the biological function of the LSD1-NEK6 CSCs in cells. The substitution of S to E is widely used to mimic the chemical properties of phosphorylated serine, but it could be an insufficient replacement and might fail to recapitulate the actual properties of a true phosphorylation[77].

Immunofluorescence of endogenous NEK6 and LSD1 showed that both proteins colocalize at distinct nuclear foci. We hypothesized that these compartments occur in part through the formation of biomolecular

condensates and first demonstrated that the IDR of LSD1 was indeed capable of forming distinct phases in vitro, which are highly dynamic and reversible (Fig. 3). Furthermore, we provided evidence that NEK6 and LSD1-IDR co-compartmentalize in vitro (Fig. 4a) and we confirmed that they also form distinct compartments in cells by endogenous protein staining (Fig. 1c) and by using the Corelet system (Fig. 5c, h)[74]. However, the Corelet CSCs established by NEK6 are more spherical and show liquid-like behavior whereas the CSCs of LSD1 are diffuse and less dynamic. We attributed this to the numerous additional contacts which LSD1 forms with nucleosomes[78], RNA[21,79], DNA[78] and other coregulator complexes[4,80,81]. Interestingly, the IDR co-partitioned specific macromolecules such as short histone peptides and DNA (Fig. 5e, f), which deviated from our expectations because structured domains of LSD1 binding histone peptides and DNA were already described[20,21,51,78,82–84]. Nevertheless, previous literature on the enzymatic activity of LSD1 suggest that there is more than the catalytic center of LSD1 where peptide substrates could bind[84]. The authors envisioned that this secondary binding site may play a role in the "stick-and-catch" model[85] where LSD1 can probe a nucleosomal substrate using its active site and secondary element to anchor the complex to remain bound to genetic loci even after catalysis was performed[84]. Furthermore, literature showed that LSD1 was unable to bind to nucleosomes in the absence of the IDR[83]. The co-partitioning of histone peptides and DNA into the IDR could be an additional mechanism of recruiting LSD1 to chromatin, which needs to be experimentally determined.

In line with the observation that the IDR can copartition macromolecules such as peptides and DNA, the IDR contributes extensively to the formation of LSD1 condensates in cells. Corelet condensates are unable to assemble when LSD1 is truncated from the N-terminus shortening its IDR (Supplementary Fig. 5a), consistent with a recent study[81]. Although the in vitro and cellular data we present provide compelling evidence that phase separation mechanisms facilitates the establishment of these localized LSD1-NEK6 CSCs, we cannot exclude the possibility that phase separation

only plays a partial role in the establishment of the LSD1-NEK6 CSC and additional mechanisms are conceivable that contribute to high concentrations of LSD1 and LSD1-related factors. To establish LSD1 CSCs reversible homo- and heterotypic interactions of the IDR could operate in conjunction with the interaction of LSD1 with chromatin and coregulator complexes[81,86].

In contrast to LSD1, the IDR of NEK6 does not seem to play any major role for the establishment of NEK6 Corelets and we did not notice any differences when the short N-terminal IDR of NEK6 is removed (Fig. 5g). Although, disordered regions show high solvent accessibility, which makes them prone to engage in non-functional interactions in contrast to globular domains[87], it was shown that also structured protein parts contribute to phase transitions[42,88,89]. On the other hand, the Corelet system, applied to provide a qualitative measure to investigate the transient LSD1-NEK6 interactions and to model CSC establishment, is artificial and may not exactly reflect the behavior of the endogenous proteins and the specific compartments. Deeper characterization of the endogenous LSD1-NEK6 CSCs using genome-wide mapping approaches such as TSA-seq[90] is needed to further assess the exact function of such compartments.

Nevertheless, the importance of the LSD1 IDR observed in the Corelet system prompted us to investigate the contribution of the IDR on gene repression mediated by LSD1 in vitro. The effect of the IDR on the activity of LSD1 has so far not been investigated in detail and previous literature only provides partial information. We observed a significant reduction of LSD1 silencing activity directly correlating with the length of N-terminal truncation (Fig. 5i). This effect could presumably be mediated by the reduced recruitment of endogenous coregulators, which are co-compartmentalized by the IDR of LSD1[81]. Previous publications showed that LSD1 activity is highly dependent on HDACs[91], CoREST[19] PRC2[22,92], and DNMT3A[80]. All of these LSD1-regulators also harbor IDRs, but their impact on the establishment and maintenance of the LSD1 microenvironment is still elusive.

In summary, our data suggest that NEK6 and LSD1 form CSCs at distinct genomic regions and that the condensation capacity of both proteins in conjunction with the phosphorylation elevates the so far well-characterized demethylation activity of LSD1. The compartmentalization could enhance local substrate concentration leading to faster demethylation rates and could also contribute to associating other chromatin binding proteins resulting in multidimensional silencing hubs (Fig. 6). Furthermore, IDRs in RNA and chromatin binding proteins can function as regulatory switches, which is frequently regulated through post-translational modifications, notably phosphorylation[81,93–98]. Studies have demonstrated that phosphorylation of the IDR in LSD1 can significantly alter specific transcription factor programs[99,100]. This suggests that the dynamic nature of the LSD1 IDR, influenced by modifications like phosphorylation, plays a crucial role in the regulation of gene expression and cellular function.

## Methods
### Plasmids
The components for the fluorescent reporter (TSECB6: pMSCV-tetO-EF1a-mCherry-2A-Blasti and rTetR-LSD1: pRRL-rTetR-LSD1-P2A-Hygro) and SGEN shRNA construct targeting LSD1 were generated in ref. 22. NEK6 shRNA guides (NEK6.1: TTGAAATACTTGATCA TCTGTG and NEK6.2: TCATCTTATCTCCATAGAAGGG) were cloned into the SGEN vector (pRRL-SFFV-GFP-miRE-PGK-NeoR), which was a kind gift from Johannes Zuber (Addgene plasmid # 111171). The Corelet constructs were kind gifts from Clifford Brangwynne (M22: pHR-SFFVp-NLS-iLID::EGFP::FTH1, Addgene plasmid # 122147, and M23: pHR-SFFVp-FUSN::mCherry::SspB, Addgene plasmid # 122148) and adapted with standard cloning techniques to exchange FUS-IDR with desired LSD1/NEK6 coding sequences. LSD1 variants and truncations were cloned into the pET28a(+) expression vector (Novagen) as C-terminal His$_6$-tag fusion and into the mammalian expression vector mVenus C1 using the Gibson Assembly method. mVenus C1 was a gift from Steven Vogel (Addgene plasmid # 27794). NEK6 coding sequence was cloned into pGEX-6P-2 as N-terminal GST-tag fusion (GE Healthcare Life Sciences).

### Antibodies
Primary antibodies used for western blot and immunohistochemistry were NEK6 (sc-D7, sc-374491; final concentration western blot 0.4 µg/ml, immunofluorescence: 0.6 µg/ml), LSD1 (Active Motif, 39186; final immunofluorescence concentration: 1.3 µg/ml) and β-ACTIN (Abcam, ab8227; final concentration 0.2 µg/ml). Secondary antibodies used for western blot were IRDye 680RD Goat anti-mouse (LI-COR) or IRDye 800CW goat anti-rabbit (LI-COR), both diluted to 0.2 µg/ml. Secondary antibodies used for immunohistochemistry were goat anti-mouse Alexa Fluor 488 (Invitrogen, #A11001), goat anti-rabbit Alexa Fluor 594 (Invitrogen, #11037), goat anti-mouse Alexa Fluor 405 (Invitrogen, # 31553) or goat anti-rabbit Alexa Fluor 405 (Invitrogen, #48258), all diluted to 1 µg/ml.

### Cell lines
NIH/3T3, HEK293, Lenti-X 293 T, and Platinum-E retroviral packaging cell lines were cultivated in DMEM high glucose media (Sigma-Aldrich) supplemented with 10% FBS, 4 mM L-glutamine, 1 mM sodium pyruvate solution, 10 mM HEPES (pH 7.3), 100 U/ml penicillin and 100 µg/ml streptomycin in an incubator providing 37 °C and 5% $CO_2$.

### Cell culture, retroviral transduction, and flow cytometry
Viral transduction was adapted from ref. 22. In brief: retroviral packaging of pMSCV vectors, 20 µg of plasmid were precipitated for 20 min in HBS buffer (140 mM NaCl, 25 mM HEPES, 0.75 mM $Na_2HPO_4$, pH 7.0) together with 125 mM $CaCl_2$ and 10 µg GagPol helper plasmid. The mix was added to a 10 cm dish with Platinum-E cells growing at 75–85% confluence in supplemented DMEM. After 16 h, the media was exchanged. Viral supernatant was gathered 40–50 h after transfection, filtered through a 0.45 µm filter, and added to the target cells at 50–70% confluence. Antibiotic selection was started two days after transduction for the reporter constructs with 10 µg/ml Blasticidin and kept up for at least 7 days. For retroviral packaging of pRRL-vectors, plasmids were mixed with helper plasmids pCMVR8.74 (pCMVR8.74 was a gift from Didier Trono, Addgene #22036) and pCAG-Eco (pCAG-Eco was a gift from Arthur Nienhuis & Patrick Salmon, Addgene #35617) and 3x w/w excess of polyethyleneimine 25 K in serum-free DMEM. The mix was added to Lenti-X cells residing in supplemented DMEM at 75-90% confluence. Media exchanges and transduction of target cells were performed as described for the reporter. Cells expressing pRRL-rTetR-LSD1-P2A-Hygro were selected with 500 µg/ml Hygromycin and cells expressing SGEN with 2.5 mg/ml Neomycin 2 days after transduction and continued for at least 7 days. Recruitment of rTetR-LSD1 was started 12 days after transduction with SGEN by treatment with 1 µg/ml Doxycycline. Expression of GFP and mCherry was analyzed using a MACSQuant Vyb flow cytometer after gating for live and single cells (Supplementary Fig. 6).

### Protein expression and purification
For His-tag purification of LSD1-His or LSD1 truncations/mutants, *E. coli BL21-CodonPlus (DE3)-RIL* cells were transformed desired plasmid and plated on LB agar with 35 µg/ml Chloramphenicol and 50 µg/ml Kanamycin and grown overnight. Subsequently, selection media were inoculated with one colony and the starter culture was cultivated at 37 °C, 150 rpm for 6 h. 500 ml LB media with respective antibiotics was inoculated with 6 ml of starter culture and cultivated at 37 °C, 150 rpm until $OD_{600}$ = 0.6. Protein expression was induced by addition of 200 µM IPTG and overexpression was performed at 17 °C, 150 rpm for 14 h. Cells were harvested at 5000 g for 15 min, 4 °C. Pellets were washed once in 30 ml STE buffer (100 mM NaCl, 10 mM Tris HCl pH 8, 1 mM EDTA) and frozen at −20 °C until use. For purification, pellets were resuspended in 30 ml sonication buffer (30 mM KPI-buffer pH 7.2, 0.2 mM DTT, 500 mM KCl, 1 mM EDTA, 10% glycerol, 20 mM imidazole) with protease inhibitor and lysed by sonication using an EpiShear Probe Sonicator (Active Motif). The lysate was cleared by centrifugation and filtration through a 0.45 µm CHROMAFIL GF/PET-45/25 filter (MACHEREY-Nagel). Affinity

chromatography was performed using an NGC™ Chromatography system (BIO-Rad) and purification columns packed with Ni-NTA superflow beads (Clontech). Proteins were eluted in elution buffer (30 mM KPI-buffer pH 7.2, 500 mM KCl, 0.2 mM DTT, 1 mM EDTA, 10% glycerol, 220 mM imidazole) and subjected to dialysis into storage buffer (20 mM HEPES pH 7.2, 200 mM KCl, 0.2 mM DTT, 1 mM EDTA, 10% glycerol). Aliquots were snap-frozen and stored at −80 °C. The same procedure was performed for respective IDR-His and IDR-mV-His proteins. GST-tagged NEK6 was expressed in a similar manner, except protein expression was induced by addition of 500 µM IPTG and further incubation at 20 °C for 14 h. Purification was performed similarly with respective adaptations for GST and buffer conditions adapted from[52], In short, cell pellets were resuspended in sonication buffer (20 mM HEPES pH 8, 500 mM MgCl2, 0.2 mM DTT, 1 mM EGTA, 10% glycerol) prior to sonication as described. Proteins bound to columns loaded with Protein Glutathione Agarose 4B beads were eluted in elution buffer (sonication buffer with 40 mM glutathione) and further dialyzed into storage buffer (20 mM HEPES pH 8, 200 mM MgCl2, 0.2 mM DTT, 1 mM EGTA, 10% glycerol).

### Western Blot
For protein expression analysis, cells were collected 13 days after transduction and antibiotic selection. Pellets were lysed in cell lysis buffer (Cell Signaling Technology®) for 30 min on ice. After 10 and 20 min of incubation, the lysate was vortexed and pipetted thoroughly to release nuclear protein. The lysate was centrifuged at $15,000 \times g$ for 10 min at 4 °C, the supernatant was mixed with 2× SDS sample buffer (125 mM Tris–HCl pH 6.8, 5% SDS, 0.004% Bromophenol Blue, 10% b-mercaptoethanol, 100 mM DTT, 20% glycerol) and heated for 10 min at 95 °C. Proteins were resolved by SDS-PAGE on a 12% polyacrylamide gel. Proteins were transferred to an Immobilon-FL PVDF membrane at 300 mA for 90 min using a wet-tank blotting system (BioRad). Proteins were detected using target specific primary antibody at manufacturer's recommendations in combination with a species-specific IRDye®800 or 680-coupled secondary antibody. Imaging was performed on an Odyssey® CLx imaging system (LI-COR).

### LSD1 activity assays
Equal amounts of recombinant His-tagged LSD1 variants were incubated at 37 °C with 12 µM H3.1 1-17 K4me1-Bt peptide (H-ARTKme1Q-TARKSTGGKAPR-Bt-NH2, Intavis) and samples were taken at distinct time points. Total peptide content was purified using self-made cleaning tips harboring C18 Cleaning Material (Agilent Technologies, Lot: 0006406886) following manufacturer's protocol for Cleanup C18 Pipette Tips (Agilent Technologies). Samples were spotted onto MALDI-TOF AnchorChip Standard target plate and covered with α-Cyano-4-hydroxycinnamic acid matrix (Sigma). The MS peak areas of monomethylated and demethylated H3K4 peptides (2181 and 2167 Da) were extracted and the fraction of demethylated peptides in the total peak area was determined to express demethylation progress during reaction time.

### NEK6 phosphorylation assay
NEK6 activity assays were adapted from ref. 52. In brief, 1.2 µM of respective phosphorylation targets were incubated for 3 h at 30 °C in phosphorylation buffer (50 mM HEPES pH 8, 10 mM MgCl2, 2.5 mM EGTA pH 8, 1 mM DTT, 0.1 mM Na3VO4, 0.1 mM PMSF and 92.5 kBq γ-32P-ATP) with 0.14 µM NEK6. Reaction was stopped by adding 2× SDS sample buffer (125 mM Tris–HCl pH 6.8, 5% SDS, 0.004% Bromophenol Blue, 10% b-mercaptoethanol, 100 mM DTT, 20% glycerol) and heated for 10 min at 60 °C. Reaction products were separated on 12% SDS-PAGE and the resulting radiation signal was detected by exposure of Storage Phosphor Screens (Molecular Dynamics) for 5–72 h. Imaging of the phosphorylation signal was performed by using the Molecular Dynamics Storm 840 Phosphor Fluorescence Scanner (Molecular Dynamics).

### In vitro droplet formation for FRAP experiments
6 µM of respective IDR-mV proteins was mixed in crowding buffer (20% PEG-8000, 40 mM TRIS-HCl pH 7.5), transferred into mPEGylated glass-bottom 384-well plate, and incubated for 2–3 h for droplet formation and settlement. Protein mobility was demonstrated by FRAP experiments performed at the Zeiss LSM980 Airyscan 2 microscope equipped with a Plan-Apochromat 63x/ 1.40 Oil DIC M27 objective in confocal mode. Bright-field and mVenus (excitation at 514 nm, detection window 517–684 nm) were imaged simultaneously every 0.5 s over 2.5 min with linear bleach event after 2 s using 514 nm laser with 40% intensity and scan speed of 8.

FRAP analysis was performed for two independent droplet formation experiments. Fluorescence signal was measured at the bleached area, at the total droplet area, and in the background as baseline signal in Zeiss ZEN software. FRAP recovery was analysed with EasyFRAP[101] using the full-scale normalization method.

### Peptide or DNA incorporation during in vitro droplet formation
For peptide or DNA incorporation, 14 µM of respective IDR-His proteins was mixed with 12 µM FITC-tagged peptide or pre-annealed 1.2 µM dsDNA labeled with Cy5 in crowding buffer (20% PEG-8000, 40 mM TRIS-HCl pH 7.5), transferred into mPEGylated glass-bottom 384-well plate and incubated for 2–3 h for droplet formation and settlement. Peptide or DNA incorporation was imaged on a Zeiss AxioObserver Spinning Disk microscope equipped with an alpha Plan-Apochromat 100x/ 1.46 Oil DIC (UV) M27 objective and an Axiocam 503 mono CCD camera.

| Peptide name | Sequence |
| --- | --- |
| H3 1-19 K9me3 | H-ARTKQTARKme3STGGKAPRK-FITC-NH2 |
| H3 1-19 | H-ARTKQTARKSTGGKAPRK-FITC-NH2 |
| H3 15-34 K27me2 | FITC-βAlaPRKQLATKAARKme2SAPATGG-NH2 |
| H3 15-34 | FITC-βAlaPRKQLATKAARKSAPATGG-NH2 |
| H3 4-19 K9me1 K14ac | Ac-KQTARKme1STGGKacAPRKQK-FITC |
| H3 1-19 S10p | FITC-βAla-ARTKQTATKpSTGGKAPRKQ-NH2 |

| Oligo name | Sequence |
| --- | --- |
| oligo 1 | *Cy5*-CGGGTTGTCAAGAATTTTAACGGCCATTTCTGTGTTGCACT CTCCTCCCGGAAG TCCCAGCTTCTGTGTTTGTGACAAACGCAAGCTCATGTAAGTGCTC |
| oligo 2 | GAGCACTTACATGAGCTTGCGTTTGTCACAAACACAGAAGCTG GGACTTCCGGGAGG AGAGTGCAACACAGAAATGGCCGTTAAAATTCTTGACAACCCG |

### Determination of turbidity, $c_{ast}$, and reversibility of in vitro droplet formation
Different amounts of IDR-mV protein were incubated for 30 min under crowding conditions (20% PEG-8000, 40 mM TRIS-HCl pH 7.5). After crowding, samples were centrifuged for 5 min at $21,500 \times g$. Supernatant was transferred and mV fluorescence was determined by using EnSpire Multimode plate reader (PerkinElmer). To determine droplet reversibility, different concentrations of KCl were added after droplet formation. Samples were mixed, centrifuged and fluorescence was measured as described. For turbidity analysis, 2 µM of respective protein was mixed under crowding conditions and transferred into glass cuvettes and OD600 was determined every minute for 30 min using Hitachi UV/Vis dual spectrometer.

### In vitro co-condensation of IDR-mV and NEK6-dsRed
2 µM of IDR-mV and NEK6-dsRed (dsRed monomer) published by Strongin et al., 2007[102] was incubated under crowding conditions for 2 h. Samples were transferred on glass slides and imaged at Zeiss CellObserver Z1 equipped with Colibri LED light source and Axiocam MRm CCD camera.

## Immunofluorescence microscopy

NIH/3T3 or HEK293 cells were cultivated until 70–90% confluency on microscopy coverslips. Cells were washed three times for 5 min with 2 ml PBS$^{Ca2+ \ Mg2+}$ (Sigma-Aldrich). Cells were fixed for 10 min at room temperature in 4% paraformaldehyde. Cells were washed as described and permeabilized with 0.2% cold TritonX-100 in PBS for 5 min. Cells were blocked in 2 ml 5% BSA in PBS$^{Ca2+ \ Mg2+}$ for 1 h at room temperature. Primary antibody binding was performed overnight at 4 °C with desired primary antibody in PBS$^{Ca2+ \ Mg2+}$/5% BSA. Secondary antibody binding was performed at room temperature for 2 h diluted in PBS$^{Ca2+ \ Mg2+}$ / 5% BSA. Cells were stained with 1 μg/ml DAPI in PBS$^{Ca2+ \ Mg2+}$ for 3 min, washed again with PBS, and mounted on microscopy slides using Mowiol® 4-88 (Sigma-Aldrich).

For 1,6-Hexanediol treatment, cells were washed as described and treated for 60 s with 10% 1,6-Hexanediol in PBS$^{Ca2+ \ Mg2+}$ prior to fixation and permeabilization.

## Investigation of corelet droplet formation in living cells

150,000 HEK293 cells were seeded into either live-cell 6-well disks or on glass slides placed into 6-well plates. After attachment, cells were transiently transfected by the use of linear polyethyleneimine 25 K with 0.8 μg M22:pHR-SFFV-NLS-iLID::EGFP::FTH1 and 3.2 μg of respective M23:pHR-SFFV-*sequence of interest*::mCherry::SspB in serum-free DMEM. For LSD1 the catalytically inactive K661A mutant was used in all Corelet droplet formation experiments. Medium was exchanged 12 h after transfection and 48–72 h after transfection, cells were further processed. For fixation, cells were placed on blue light-emitting LED panel for 4 h with on/off cycle of 5 min with cell fixation under blue light for 10 min by use of 4% paraformaldehyde followed by immunostaining as explained in "Immunofluorescence microscopy".

## Cell image acquisition

For colocalization imaging of fixed cells, samples were analyzed on Zeiss LSM710 equipped with a Plan-Apochromat 63x/ 1.4 Oil DIC objective. Lasers and detection wavelength were selected to exclude cross-detection. Z-stacks covering the whole nucleus were acquired applying an interval of 440 nm and single slices were selected. For live-cell droplet formation experiments, cells were imaged at the LSM980 Airyscan 2 microscope equipped with a Plan-Apochromat 63x/ 1.4 Oil DIC objective in confocal mode. First, cells were imaged in mCherry channel for 5 s to gather pre-induction images. Second, droplet formation was induced for a period of 20 s of dual imaging of mCherry and EGFP with continuous 488 nm laser activation using 5% intensity with 5 frames/s. Dissolution of biomolecular condensates was imaged only via mCherry detection with a time interval of 2 s.

## Mass spectrometry

Protein samples originating from the phosphorylation assay of LSD1+/− NEK6 were purified following the SP3 protocol[103], reduced and alkylated with TCEP and CAA, and digested with either Trypsin, GluC or ProAlanase. Peptide LC-MS/MS analysis was performed in four replicates by using an EASY-nLC 1200 (Thermo Fisher) coupled to an Exploris 480 mass spectrometer (Thermo Fisher)[104]. Separation of peptides was performed on 25 cm fritted silica emitters (CoAnn Technologies, 0.75 μm inner diameter), packed in-house with reversed-phase ReproSil-Pur C$_{18}$ AQ 1.9 μm resin (Dr. Maisch). The column was constantly kept at 50 °C. Peptides were eluted in 12 min applying a segmented linear gradient of 0% to 98% solvent B (solvent A 0% ACN, 0.1% FA; solvent B 80% ACN, 0.1% FA) at a flow rate of 300 nL/min[105]. Mass spectra were acquired in data-dependent acquisition mode. MS$^1$ scans were acquired at an Orbitrap resolution of 120,000 with a scan range (m/z) of 280–1500, a maximum injection time of 100 ms, and a normalized AGC target of 300%. For fragmentation, only precursors with charge states 2–6 were considered. Up to 20 Dependent Scans were taken. For dynamic exclusion, the exclusion duration was set to 40 s and a mass tolerance of +/− 10 ppm. The isolation window was set to 1.6 m/z with no offset. A normalized collision energy of 30 was used. MS$^2$ scans were taken at an Orbitrap resolution of 15,000, with a fixed first mass (m/z) = 100. Maximum injection time was 150 ms and the normalized AGC target 5%.

Raw data was analyzed with MaxQuant 2.0.3.0 using default settings[106], besides the following adjustments: A fasta-file containing the *E. coli* reference proteome (downloaded from uniport.org 2021-09-03) was used and the sequences of the expression constructs of LSD1 and Nek6 were added. Phosphorylation of STY was set as variable modification and separate parameter groups were used for runs originating from different digests with the respective peptidase set. Match between runs was activated.

## Statistics and reproducibility

Statistical analyses were performed using Prism 8.0 (GraphPad software). Detailed statistical methods are described in respective figure legends. Unless mentioned otherwise, all experiments were performed three independent times. For comparison of two individual groups, a Student's t-test was performed. For comparison of more than two groups, a one-way ANOVA with post-hoc test was performed. For each experiment, differences were considered significant for $*p < 0.05$, $**p < 0.01$, $***p < 0.001$, $****p < 0.0001$.

## Reporting summary

Further information on research design is available in the Nature Portfolio Reporting Summary linked to this article.

## Data availability

Raw data and search result files are available via the JPost repository[107] under URL: https://repository.jpostdb.org/entry/JPST002480. The numerical source data for all main and supplementary graphs are provided as Excel file named Supplementary Data. All uncropped pictures are shown in Supp. Figs. 7 and 8.

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

## Acknowledgements

We thank all former and current members of the Rathert and Jeltsch lab for reagents, protocols, and discussions and Regina Philipp for technical support. We thank A. Jeltsch (University of Stuttgart) for insightful comments and discussion on the manuscript. We gratefully acknowledge the Technology Platform "Cellular Analytics" of the Stuttgart Research Center Systems Biology (SRCSB) for their support & assistance in this work, as well as DFG for instrumentation (INST 211/744-1 FUGG). We would like to thank Paulina Heinkow for technical assistance, especially in maintaining the LC-MS/MS instruments at the MSPUB. The work described here was supported by the Wilhelm-Sander Foundation (2016.082.1, 2020.055.1) and the German Cancer Aid (70113426, 70113433).

## Author contributions

F.K. and P.R. conceived the study and designed experiments. F.K. performed and analyzed most experiments, J.E. performed the LC-MS/MS experiment and analysis supervised by I.F., S.E. provided assistance with microscopy, P.R. analyzed data and supervised the research. F.K. and P.R. co-wrote the paper. All authors have read and agreed to the published version of the manuscript.

## Funding

## Competing interests

The authors declare no competing interests.
