## [Transparent Peer Review file · Communications Biology]

The kinase NEK6 positively regulates LSD1 activity and accumulation in local chromatin sub-compartments

Corresponding Author: Dr Philipp Rathert

Version 0:

Reviewer comments:

Reviewer #1

(Remarks to the Author)

In this paper, the authors have investigated the role of two chromatin regulators, NEK6 and LSD1, in the formation of phase-separated condensates in the nucleus called chromatin sub-compartments (CSCs). Posttranslational modifications tune the phase behavior of phase separating proteins and consequently the biological functions of the condensates that contain them. The authors show that the Ser/Thr kinase NEK6 phosphorylates a conserved serine residue within the intrinsically-disordered region (IDR) of the histone lysine demethylase LSD1. The observations, many of which are counterintuitive, raise interesting questions regarding the IDR function in chromatin subcompartments and the complex regulation of LSD1 activity in cellular processes like cell cycle. This work is an important reminder that associations between IDRs, phase separation, and biological function may not always be mechanistically straightforward and generalizable. The findings are relevant to the fields of chromatin biology and nuclear condensates, and highlight the complex interplay of biochemical activity, condensate formation, and function. However, there are several issues/concerns that need to be addressed before I can recommend publication.

1. The premise for connecting LSD1 and NEK6 is unclear from the introduction. The authors can discuss the results from the genetic screening in their earlier report. They should also introduce chromatin subcompartments in the introduction and highlight the association with LSD1 activity and mitosis. The term "CSC" is introduced before expanding the acronym in the last line of the introduction.
2. The multiple sequence alignment shown in Fig. 2f is of little value since every residue is 100% identical. The authors should perform an alignment of more phylogenetically distant sequences.
3. It would be interesting to check using Western blotting whether there is an accumulation of phosphorylated LSD1 in mitotically-arrested cells.
4. The observation that LSD1 IDR-mV droplets are insensitive to salt concentrations raises questions regarding the interactions that drive phase separation. Does pH changes affect the droplets?
5. The FRAP recovery rates in Fig. 3e have not been quantified. The droplet coalescence images are not convincing as one of the two droplets look like air bubble to me. Is it possible to show another field of view?
6. It is unclear to me what the turbidity kinetics say about the phase-separation propensities of the WT vs. S126E mutant. The authors have not shown any evidence for reversibility upon changing physical conditions. Overall, I am not convinced with the phase-separation data and would encourage the authors to delve deeper into the phase behavior of LSD1-IDR. At a minimum, one needs to observe a saturation concentration (c_{sat}) and reversibility. If droplet formation is irreversible, the authors may want to comment on the physical and biological implications.
7. In suppl. Fig. 3c, the authors show FRAP curves of WT and S126E. Although S126E is a phosphomimetic mutant, claiming that phosphorylation did not alter mobility is misleading since it implies that the experiment was performed with in vitro phosphorylated WT protein.
8. To ensure that the observed effects of substituting Ser126 with Glu stem solely from the introduction of a negative charge

and not from the removal of the Ser side chain, the authors should introduce a Ser126Ala mutation. A preliminary characterization of its phase separation behavior under conditions where the WT forms droplets in vitro would be informative. If the Ser126Ala mutant also exhibits phase separation, it would suggest that the negative charge introduced by Glu is the primary driver of the observed effects. Conversely, if the Ser126Ala mutant fails to undergo phase separation, it would indicate that the Ser side chain itself plays a role in phase separation.

9. In line 191, why do the authors specify that the heterotypic interactions between LSD1 IDR and NEK6 are "weak"?

10. The observation of aggregates upon expression of NEK6-dsRed may be attributed to the tendency of dsRed to form tetramers. Since the authors do use mVenus for LSD1, is there a particular reason for using dsRed instead of mRuby or other monomeric fluorescent proteins?

11. As the authors rightfully point out in line 196 that the use of 1,6-hexanediol in phase separation studies has issues, why use it?

12. I congratulate the authors for their attempts at applying the Corelet system to monitor LSD1 condensates in a spatiotemporally regulated manner, but I am not sure if the Corelet system offers new insights on the biology of CSCs. Is it possible to dissect the role of the folded domains of LSD1 in mediating multivalent interactions that are presumably critical for phase separation? Is the demethylase activity important for condensate formation?

13. I would have liked to have seen the data (not shown) on the cellular toxicity of LSD1 overexpression.

14. The two channels in Fig. 4f have not been labeled. The false coloring is also the same in both channels.

15. How do the authors know whether the LSD1 or NEK6 Corelets are indeed CSCs and not unnatural condensates?

General comment:

The overall conclusion from the individual sections are not clear. A few sentences to summarize the findings will help improve understanding since the observations on the in vivo role of LSD1 IDR are confusing, which is probably indicative of the complex interplay of intramolecular interactions, LSD1-NEK6 interactions, and catalytic activity in LSD1 biology.

Reviewer #2

(Remarks to the Author)

This manuscript by Franziska, Philipp, and colleagues presents a thorough and rigorous investigation demonstrating that LSD1 and NEK6 colocalize both in vivo and in vitro. They show that the intrinsically disordered N-terminal of LSD1 can form phase separation, which is crucial for LSD1 activity. The experimental design is logical, the results are clearly presented, and the conclusions are well supported. The only criticism might be whether the manuscript offers results of sufficient general interest to warrant publication in Communications Biology. However, I believe it does, as the findings pertain to LSD1-mediated cellular processes and, consequently, to gene regulation mechanisms and other DNA transactions regulated by LSD1, which are of broad interest.

There is little to criticize here, although it would be nice to expand the discussion of IDR of LSD1, for example, what's the amino acid composition of LSD1 N-terminal, it seems it's highly negatively charged. Is there any other data supporting the IDR conclusion, like CD or NMR? Have you tried made a new LSD1 mutant in which the primary amino acid sequence of N-terminal was scrambled but the amino acid composition keep unchanged? It would be interesting to see if this mutant affect LSD1 activity.

Supplemental Fig 1a, "wells were harvested" should be "cells were harvested"

Supplemental Fig 1b, the quality of immunofluorescence microscopy images is not good, LSD1 and NEK6 can not be seen in the image.

Supplemental Fig 2a, there are multiple bands for the LSD1 lane, which band is the expected full-length LSD1? What are other bands, degraded LSD1?

For the in vitro assay used 2-10uM IDR-mV, is this concentration physiologically relevant?

Line 288, "supplementary Fig. 3e" should be "3f".

Version 1:

Reviewer comments:

Reviewer #1

(Remarks to the Author)

There is a minor grammatical error in the revised Fig 4. The text in panel D should be "Box plots depicting fluorophore" instead of "Box plots depicting of fluorophore".

Other than that, the authors have satisfactorily addressed all my concerns and the revisions have significantly improved the manuscript in my opinion.

Reviewer #2

(Remarks to the Author)

All my comments have been addressed.

Point-by-point reply to the referee's comments

Reply for the reviewers: We would like to thank all reviewers for their work and fairness in reviewing our manuscript. Based on the comments by the reviewers, we have conducted several additional experiments and provided the requested controls. Moreover, we have refined our manuscript and included the suggestions of the reviewers. These changes improved the manuscript significantly.

We are very thankful to all two reviewers for their fair and constructive criticism in this round of revision.

Summary of the novel experiments and additional figures, which are included in the revised manuscript:

- **Western Blots** showing validation of S126p-specific antibody (reviewer figure 1)
- **Droplet formation under different pH conditions:** Fig 3e
- **Comparison between IDR-WT, IDR-S126A and IDR-S126E:** Supplementary Fig. 3f

Reviewer 1, Point 6: Saturation and reversibility:

- Representative pictures demonstrating **reversibility** of IDR-mV droplet formation upon addition of salt: Fig. 3h
- Detailed **comparison of the droplet reversibility** with addition of salt between IDR-mV WT, S126A and S126A mutant: Fig. 4g
- Determination of **saturation concentration** (C_{sat}) of IDR-mV WT, S126A and S126E: Fig 4h
- **Corelet droplet reversibility** upon blue light removal: Supplementary videos 8 and 9

Reviewer 1, Point 7: FRAP curves and difference between S126p and S126E:

- Detailed analysis of FRAP curves including recovery half time and mobile fraction: Fig. 4c, d
- We highlight the use of the phosphomimic variant S126E throughout the manuscript and discuss it in detail in the discussion.
- **Revised sequence alignment:** Fig. 2f
- **CD spectra of IDR-LSD1** in comparison to secondary structure elements: Fig 3b
- Amino acid occurrence of IDR-LSD1 with chemical properties
- **Revised droplet coalescence demonstration:** additional fusion event and droplet size analysis before and after fusion: Fig 3f, g; supplementary video 3

Reviewers' comments:

Reviewer #1 (Remarks to the Author):

In this paper, the authors have investigated the role of two chromatin regulators. NEK6 and LSD1, in the formation of phase-separated condensates in the nucleus called chromatin sub-compartments (CSCs). Posttranslational modifications tune the phase behavior of phase separating proteins and consequently the biological functions of the condensates that contain them. The authors show that the Ser/Thr kinase NEK6 phosphorylates a conserved serine residue within the intrinsically-disordered region (IDR) of the histone lysine demethylase LSD1. The observations, many of which are counterintuitive, raise interesting questions regarding the IDR function in chromatin subcompartments and the complex

regulation of LSD1 activity in cellular processes like cell cycle. This work is an important reminder that associations between IDRs, phase separation, and biological function may not always be mechanistically straightforward and generalizable. The findings are relevant to the fields of chromatin biology and nuclear condensates, and highlight the complex interplay of biochemical activity, condensate formation, and function. However, there are several issues/concerns that need to be addressed before I can recommend publication.

Reply: We thank the referee for highlighting the novelty and relevance of our work. As suggested by the reviewer, we have collected additional data in this revision to further support our initial hypothesis and to address the reviewer's concerns. (see below).

1. The premise for connecting LSD1 and NEK6 is unclear from the introduction. The authors can discuss the results from the genetic screening in their earlier report.

Reply: We thank the reviewer for bringing this to our attention. This has now been clarified in the introduction.

They should also introduce chromatin subcompartments in the introduction and highlight the association with LSD1 activity and mitosis.

Reply: We have now included a section in the introduction, where we introduce chromatin sub-compartments in more detail. Furthermore, we also include a short section for the function of LSD1 in mitosis.

The term "CSC" is introduced before expanding the acronym in the last line of the introduction.

Reply: We thank the reviewer for highlighting this mistake and corrected this.

2. The multiple sequence alignment shown in Fig. 2f is of little value since every residue is 100% identical. The authors should perform an alignment of more phylogenetically distant sequences.

*Reply: We understand the criticism and now include additional distant species such as *Pelodiscus sinensis* (Chinese softshell turtle, PELSI), *Danio rerio* (Zebrafish, DANRE) and *Xenopus laevis laevis* (XENLA). However, all additional vertebrates containing the respective region show 100% sequence similarity, demonstrating that this region is very important. Other, non-vertebrate organisms such as *C. elegans* and *D. melanogaster* for example do not contain the identified sequence, which suggest that this regulatory mechanism occurred at later evolutionary stages. We now comment on this in the revised manuscript.*

3. It would be interesting to check using Western blotting whether there is an accumulation of phosphorylated LSD1 in mitotically-arrested cells.

Reply: We have raised an antibody against phosphorylated LSD1, which works well with purified proteins in a western blot (Reviewer figure 1a). Unfortunately, we were unable to obtain any meaningful results in western blots from cell lysates and immunofluorescence (Reviewer figure 1b-c). Therefore, we are unable to provide this data.

Reviewer figure 1: Validation of LSD1-S126p specific antibody.

a) The LSD1 S126p antibody reveals a strong band at respective height for in vitro phosphorylated LSD1: FL-LSD1 WT or S126A was incubated with or without NEK6 and later dissolved on SDS-PAGE and transferred as western blot for antibody validation (top). The same sample was dissolved on SDS-PAGE and stained with Coomassie as loading control (bottom).

b) Western blot loaded with whole cell protein lysate reveals smeary signal without clear band at respective height. NIH/3T3 cells expressing rTetR-LSD1 were harvested, lysed and loaded on two SDS-PAGE lanes for western blotting to detect LSD1 and phosphorylated LSD1.

c) Representative immunofluorescence microscopy images of fixed NIH/3T3 cells stained with anti-NEK6 and anti-LSD1-S126p reveals unspecific binding of the generated antibody. Scale bar = 10 μ m.

4. The observation that LSD1 IDR-mV droplets are insensitive to salt concentrations raises questions regarding the interactions that drive phase separation. Does pH changes affect the droplets?

Reply: We speculate that indeed higher salt concentrations might change droplet formation, but in the range that we tested we observed droplet formation in all cases, even though the droplets at 100 mM KCl seem to be smaller than the ones at 30 mM KCl. However, we never quantified this observation. In line with this argument, we also observe that an increase in KCl after the droplet formation was initiated leads to a dissolution of the formed droplets. We now include experiments investigating the effect of the pH on droplet formation of the IDR in the revised manuscript showing that indeed the condensation of LSD1-IDR seems to be dependent on the pH of the buffer. We observe fiber-like condensates or aggregates at pH 7, regular shaped droplets at pH 7.5, a striking reduction in condensate size and number at pH 8 and no meaningful droplets at pH 9 (Figure 3e).

5. The FRAP recovery rates in Fig. 3e have not been quantified.

Reply: We showed the FRAP analysis in Supplementary figure 3c. This data demonstrates dynamic protein fractions within the droplets of both, WT and S126E LSD1-IDR. But we did not observe a striking difference between WT and S126E IDR. However, we now include the FRAP curves together with a more detailed analysis in the main figure (Fig. 4b-d) including recovery half time and mobile fraction for each protein. The data shows a large variance, but we observe a slower recovery and mildly elevated mobile fraction for the S126E phosphomimic variant. Based on this data, we speculate that phosphorylation of S126, mimicked by the exchange of S126 to E, affects the phase separation, which is reflected by bigger droplets, stronger and faster increase in turbidity and a higher mobile fraction as well as a reduced droplet aging.

The droplet coalescence images are not convincing as one of the two droplets look like air bubble to me. Is it possible to show another field of view?

Reply: We have to politely disagree with the reviewer. In Figure 3d (now Fig. 3f) we show snapshots of the provided videos. In the images and videos it is clearly visible that after the fusion event the droplet in the field of view increases in size. Since these are snapshots from a time series imaged without z-stacks, we cannot show another section. However, we have included one additional coalescence event with another droplet as well as the corresponding video (supplementary video 3) and quantified the area of the droplets before and after coalescence (Fig. 3f, g).

6. It is unclear to me what the turbidity kinetics say about the phase- separation propensities of the WT vs. S126E mutant.

Reply: During the crowding experiment, droplets start to form and we observe that the phosphomimic S126E variant shows a faster increase in turbidity than the WT IDR, which suggests that larger droplets are formed faster compared to the WT. A faster increase in turbidity suggests that the mutant protein is more prone to phase separating into dense liquid-like or solid-like phases. However, this is counter intuitive in our case since the exchange of S126 to E does not seem to affect the saturation concentration (Fig. 4h). Nevertheless, the turbidity result is confirmed by the observation that droplets formed by the S126E variant have a higher mobile fraction and are larger in size and intensity (Fig. 4b, d-f).

The authors have not shown any evidence for reversibility upon changing physical conditions. Overall, I am not convinced with the phase-separation data and would encourage the authors to delve deeper into the phase behavior of LSD1-IDR. At a minimum, one needs to observe a saturation concentration (c_{sat}) and reversibility. If droplet formation is irreversible, the authors may want to comment on the physical and biological implications.

Reply: We now provide additional experiments where we determine c_{sat} (Fig. 4h) and demonstrate condensation reversibility (Figure 3h, 4g). These results suggest that although we do observe a faster formation of droplets, larger and more intense droplets and a higher mobile fraction within the condensate this is not reflected in a lower c_{sat} . However, we do observe that the S126E variant shows a better droplet dissolution upon treatment with high salt concentrations indicating that the droplets show reduced condensate aging.

7. In suppl. Fig. 3c, the authors show FRAP curves of WT and S126E. Although S126E is a

phosphomimetic mutant, claiming that phosphorylation did not alter mobility is misleading since it implies that the experiment was performed with in vitro phosphorylated WT protein.

Reply: We agree with the reviewer that the E-mutant is not an exact mimic of a phosphorylated serine and have now included this information in the discussion. We tried to perform in vitro phosphorylation prior FRAP, but were unsuccessful to retrieve enough pure IDR for FRAP experiments (data not shown) and purification would be necessary due to the strong co-compartmentalization behaviour of NEK6. Also, co-expression of NEK6 and LSD1-IDR in E.coli followed by purification was unsuccessful leading to co-purification of NEK6 (data not shown).

8. To ensure that the observed effects of substituting Ser126 with Glu stem solely from the introduction of a negative charge and not from the removal of the Ser side chain, the authors should introduce a Ser126Ala mutation. A preliminary characterization of its phase separation behavior under conditions where the WT forms droplets in vitro would be informative. If the Ser126Ala mutant also exhibits phase separation, it would suggest that the negative charge introduced by Glu is the primary driver of the observed effects. Conversely, if the Ser126Ala mutant fails to undergo phase separation, it would indicate that the Ser side chain itself plays a role in phase separation.

Reply: We now provide evidence that also the S126A mutant shows phase behaviour to a similar extent as the WT (Fi. 4g,h and Supplementary Fig. 3f).

9. In line 191, why do the authors specify that the heterotypic interactions between LSD1 IDR and NEK6 are "weak"?

Reply: We apologize for the confusion, but this was non-validated interpretation, as phase separation interactions are mostly defined as weak heterotypic interactions. We removed this in the revised manuscript.

10. The observation of aggregates upon expression of NEK6-dsRed may be attributed to the tendency of dsRed to form tetramers. Since the authors do use mVenus for LSD1, is there a particular reason for using dsRed instead of mRuby or other monomeric fluorescent proteins?

Reply: We are very sorry that we did not clarify this in detail. We used the DsRed monomer published by Strongin et al., 2007¹. We highlight this in the materials and methods now.

11. As the authors rightfully point out in line 196 that the use of 1,6-hexanediol in phase separation studies has issues, why use it?

Reply: Hexanediol was initially used at the start of the project to determine if phase separation mechanisms drive the interaction between endogenous LSD1 and NEK6. This was at a time where Hexanediol was still used by numerous groups and the criticism about additional artefacts arose later^{2,3}. However, we still believe that these results should be included in the manuscript since they contribute to the experimental rationale.

12. I congratulate the authors for their attempts at applying the Corelet system to monitor LSD1 condensates in a spatiotemporally regulated manner, but I am not sure if the Corelet system offers new insights on the biology of CSCs. Is it possible to dissect the role of the folded domains of LSD1 in mediating multivalent interactions that are presumably critical for phase separation?

Reply: We used the Corelet system to demonstrate condensation ability of both proteins in living cells, as the immunofluorescence data was performed in fixed cells and with Hexandiol treatment, which is seen critical in the field. Using the Corelet system we observe that LSD1 requires the folded domains and the IDR to form condensates (Fig. 5 and Supplementary Fig. 5). Neither the IDR, nor the folded domains alone show the formation of Corelets. We strongly believe that the Corelet system offers new insights on the biology of LSD1-NEK6 CSCs since we observe that both, the NEK6 and the LSD1 Corelets incorporate the respective endogenous partner (Fig. 5h).

Is the demethylase activity important for condensate formation?

Due to this toxicity observed with WT LSD1, all Corelet experiments were performed with the K661A mutant, which has a strongly reduced demethylase activity and we clarified this in the revised manuscript. This suggests that the demethylase activity might not play such an important role. However, we cannot exclude this, because we could not compare this to the wt protein. Reviewer Fig. 2 shows data demonstrating that high LSD1 expression is toxic leading to a large number of detached and rounded HEK293 cells transfected with wtLSD1. This is not apparent when the K661A-LSD1 mutant is transfected.

Reviewer Figure 2: Transient transfection of LSD1 WT is toxic for HEK293 cells.
Representative bright field pictures of HEK293 cells 48 h after transfection of either WT or catalytic inactive LSD1 (K661A) fused to mVenus (not shown). Scale bar = 400 μ m.

13. I would have liked to have seen the data (not shown) on the cellular toxicity of LSD1 overexpression.

Reply: We now provide the requested data in Reviewer figure 2. The explanation is provided in the comment above.

14. The two channels in Fig. 4f have not been labeled. The false coloring is also the same in both channels.

Reply: This seems to be a misunderstanding. Figure 4f (now figure 5f) shows two representative images of two independent experiments of phase-separated IDR-LSD1 with incorporated Cy5-labelled DNA. We improved the labelling together with Figure 4e (now figure 5e) to highlight that we show phase separated IDR-LSD1 with co-compartmentalization of histone peptides (4e, now 5e) and dsDNA (4f, now 5f).

15. How do the authors know whether the LSD1 or NEK6 Corelets are indeed CSCs and not unnatural condensates?

Reply: We agree with reviewer 1 and admit that we had similar concerns. However, we see striking incorporation of endogenous LSD1 in NEK6 Corelets and vice versa, which was not shown for both endogenous proteins in FUS Corelets, indicating that at least to some extent the NEK6/LSD1 Corelets are close to the CSCs we observe in the immunofluorescence images of the endogenous proteins. Further characterization of the endogenous CSCs by TSA-seq⁴ and TSA-MS⁵ and a direct comparison with the Corelets would clarify in more detail if they resemble a true copy of the endogenous CSC, which would go beyond the presented study.

General comment:

The overall conclusion from the individual sections are not clear. A few sentences to summarize the findings will help improve understanding since the observations on the in vivo role of LSD1 IDR are confusing, which is probably indicative of the complex interplay of intramolecular interactions, LSD1-NEK6 interactions, and catalytic activity in LSD1 biology.

Reply: We thank the reviewer for this feedback. While the nature of the biology does present some inherent complexity, we have taken care to present the findings as clearly as possible and revised the manuscript.

Reviewer #2 (Remarks to the Author):

This manuscript by Franziska, Philipp, and colleagues presents a thorough and rigorous investigation demonstrating that LSD1 and NEK6 colocalize both in vivo and in vitro. They show that the intrinsically disordered N-terminal of LSD1 can form phase separation, which is crucial for LSD1 activity. The experimental design is logical, the results are clearly presented, and the conclusions are well supported. The only criticism might be whether the manuscript offers results of sufficient general interest to warrant publication in Communications Biology. However, I believe it does, as the findings pertain to LSD1-mediated cellular processes and, consequently, to gene regulation mechanisms and other DNA transactions regulated by LSD1, which are of broad interest.

Reply: We thank the reviewer for the conclusion that our manuscript provides data which are of broad interest and thereby “warrant publication in Communications Biology”. We have added additional data and revised the manuscript in order to match the reviewer’s concerns, which are outlined below.

There is little to criticize here, although it would be nice to expand the discussion of IDR of LSD1, for example, what’s the amino acid composition of LSD1 N-terminal, it seems it’s highly negatively charged. Is there any other data supporting the IDR conclusion, like CD or NMR?

Reply: We now provide an additional figure showing the amino acid composition of the IDR as well as the distribution of the individual amino acids (Supplementary Fig. 3b). This shows that the IDR has a negative net charge and contains a large number of hydrophobic AAs, as well as several acidic and basic patches. Furthermore, we show a CD measurement of the IDR, which demonstrates that the protein is unfolded compared to CD reference data of structural elements ⁶ (Fig. 3b).

Have you tried made a new LSD1 mutant in which the primary amino acid sequence of N-terminal was scrambled but the amino acid composition keep unchanged? It would be interesting to see if this mutant affect LSD1 activity.

Reply: We agree with reviewer 2 that this would be an interesting experiment, which we have planned for the future together with selective aa exchanges. However, the current manuscript aims to investigate the connection between LSD1 and NEK6 and the role of the phosphorylation catalyzed by NEK6. A randomized AA sequence of the IDR would be informative only with additional exchanges of all potential candidate AAs contributing to the phase separation behaviour of the LSD1 IDR, such as lysine, arginine, serine and threonine residues. This would go beyond and distract from the primary goal of this study.

Supplemental Fig 1a, “wells were harvested” should be “cells were harvested”

Reply: Thank you or highlighting this mistake, we corrected it in the revised manuscript.

Supplemental Fig 1b, the quality of immunofluorescence microscopy images is not good, LSD1 and NEK6 can not be seen in the image.

Reply: We have kept the intensities of all immunofluorescence images the same in all channels intentionally. We agree that it is difficult to see LSD1 and NEK6 signal with these settings, which already indicates the dispersed localization of both proteins and no

connection to condensed chromatin during mitosis. We now provide the same images (Supplementary Figure 1c) with enhanced intensities demonstrating that LSD1 and NEK6 are broadly distributed in mitotic cells, are not associated with the chromosomes and show no colocalization.

Supplemental Fig 2a, there are multiple bands for the LSD1 lane, which band is the expected full-length LSD1? What are other bands, degraded LSD1?

Reply: We Thank the reviewer for highlighting this. We now marked the band corresponding to LSD1-His and all other purified proteins in the figures, whenever more bands were visible in respective gel picture. The additional bands often occur in His-purifications, especially when purifying bigger proteins. The band above 70 kDa is a chaperone and all additional bands are most likely co-purified proteins with weak affinity towards Ni-NTA beads.

For the in vitro assay used 2-10uM IDR-mV, is this concentration physiologically relevant?

Reply: We agree with reviewer 2 that a concentration of 2-10 μ M is far above the concentration of the endogenous protein. Even the concentration of a very abundant protein such as GAPDH is below 0.5 μ M⁷. However, it is impossible to match the conditions of a living cell in a tube, where thousands of additional biological molecules influence protein function. In addition, this is a concentration range on the lower end currently used in the field. Most groups are working with much higher concentrations, see⁸⁻¹¹ for example, and we need to find a compromise between detectability of an effect and physiologic conditions as close as possible to a cellular system.

Line 288, "supplementary Fig. 3e" should be "3f".

Reply: We thank the reviewer for highlighting this and corrected the mistake.

Literature

- 1 Strongin, D. E. *et al.* Structural rearrangements near the chromophore influence the maturation speed and brightness of DsRed variants. *Protein Engineering, Design and Selection* **20**, 525-534 (2007). <https://doi.org:10.1093/protein/gzm046>
- 2 Kroschwald, S., Maharana, S. & Simon, A. W.
- 3 Düster, R., Kaltheuner, I. H., Schmitz, M. & Geyer, M. 1,6-Hexanediol, commonly used to dissolve liquid-liquid phase separated condensates, directly impairs kinase and phosphatase activities. *J Biol Chem* **296**, 100260 (2021). <https://doi.org:10.1016/j.jbc.2021.100260>
- 4 Dopie, J., Sweredoski, M. J., Moradian, A. & Belmont, A. S. Tyramide signal amplification mass spectrometry (TSA-MS) ratio identifies nuclear speckle proteins. *J Cell Biol* **219** (2020). <https://doi.org:10.1083/jcb.201910207>
- 5 Zhang, L. *et al.* TSA-seq reveals a largely conserved genome organization relative to nuclear speckles with small position changes tightly correlated with gene expression changes. *Genome Res* **31**, 251-264 (2021). <https://doi.org:10.1101/gr.266239.120>
- 6 Yang, J. T., Wu, C.-S. C. & Martinez, H. M. in *Methods in Enzymology* Vol. 130 208-269 (Academic Press, 1986).
- 7 Lazarev, V. F., Guzhova, I. V. & Margulis, B. A. Glyceraldehyde-3-phosphate Dehydrogenase is a Multifaceted Therapeutic Target. *Pharmaceutics* **12** (2020). <https://doi.org:10.3390/pharmaceutics12050416>

- 8 Bremer, A. *et al.* Deciphering how naturally occurring sequence features impact the phase behaviours of disordered prion-like domains. *Nat Chem* **14**, 196-207 (2022). <https://doi.org:10.1038/s41557-021-00840-w>
- 9 Wang, J. *et al.* A Molecular Grammar Governing the Driving Forces for Phase Separation of Prion-like RNA Binding Proteins. *Cell* **174**, 688-699.e616 (2018). <https://doi.org:10.1016/j.cell.2018.06.006>
- 10 Martin, E. W. *et al.* Interplay of folded domains and the disordered low-complexity domain in mediating hnRNPA1 phase separation. *Nucleic Acids Research* **49**, 2931-2945 (2021). <https://doi.org:10.1093/nar/gkab063>
- 11 Keenen, M. M. *et al.* HP1 proteins compact DNA into mechanically and positionally stable phase separated domains. *Elife* **10** (2021). <https://doi.org:10.7554/eLife.64563>